# Magmatism controls global oceanic transform fault topography

Xiaochuan Tian [1] ✉, Mark D. Behn [1], Garrett Ito[2], Jana C. Schierjott [2], Boris J. P. Kaus [3] & Anton A. Popov[3]

Oceanic transform faults play an essential role in plate tectonics. Yet to date, there is no unifying explanation for the global trend in broad-scale transform fault topography, ranging from deep valleys to shallow topographic highs. Using three-dimensional numerical models, we find that spreading-rate dependent magmatism within the transform domain exerts a first-order control on the observed spectrum of transform fault depths. Low-rate magmatism results in deep transform valleys caused by transform-parallel tectonic stretching; intermediate-rate magmatism fully accommodates far-field stretching, but strike-slip motion induces across-transform tension, producing transform strength dependent shallow valleys; high-rate magmatism produces elevated transform zones due to local compression. Our models also address the observation that fracture zones are consistently shallower than their adjacent transform fault zones. These results suggest that plate motion change is not a necessary condition for reproducing oceanic transform topography and that oceanic transform faults are not simple conservative strike-slip plate boundaries.

Oceanic transform faults display a wide range of topographic morphologies[1,2]. Overall, the broad-scale topographies of transform faults deepen systematically with decreasing spreading rate[3,4] (Fig. 1d and Supplementary Fig. 1g). At the fastest seafloor spreading rates, the transform zone is relatively shallow, sometimes even shallower than the adjacent seafloor (Mode 1, Fig. 1a and Supplementary Fig. 1a, b). At fast to intermediate spreading rates, shallow valleys with less than 1 km of relief delineate the fault zone (Mode 2, Fig. 1b and Supplementary Fig. 1c, d). Finally, at slow to ultra-slow spreading ridges, deep (>1 km of relief) transform valleys form and the corresponding fracture zones are deeper than the adjacent seafloor (Mode 3, Fig. 1c and Supplementary Fig. 1e, f). Note that while we broadly categorize transform topography into three discrete modes, these variations with spreading rate occur on a continuum. Moreover, other processes may be at play for generating complex smaller-scale, time-dependent morphologies[2]. Another global observation is that fracture zones are consistently shallower than their adjacent transforms by an average of ~650 m[3]. Although a systematic relationship between spreading rate and the

morphology of mid-ocean ridges is well documented and understood to be related to differences in magma supply[5,6], the cause of the spectrum in oceanic transform topography, and its contrast with the adjacent fracture zones, is still not clear ~60 years after oceanic transform faults and fracture zones were first discovered[7,8].

A few mechanisms are frequently invoked to explain transform fault topography. The first suggests that plate motion changes generate transform-perpendicular compression or extension leading to the formation of transform ridges or valleys, respectively[9,10]. However, plate motion changes are not persistent in time (see ref. 2 and references therein) and some transform ridges and valleys arise without significant plate motion changes[11], suggesting other fundamental causes. An alternative model invokes the nonlinear viscoelastic response due to the shearing of two adjacent plates to explain the observed transverse ridges that bound the transform valleys[12]. This model relates transform valley depth to fault shear stress. However, the two mechanisms above do not explain the observed spreading rate dependence of transform valley depth, nor the depth difference

---

[1]Department of Earth and Environmental Sciences, Boston College, Chestnut Hill, MA, USA. [2]Department of Earth Sciences, University of Hawaii, Honolulu HI, USA. [3]Institute of Geosciences, Johannes Gutenberg University Mainz, Mainz, Germany. ✉e-mail: x.tian@bc.edu

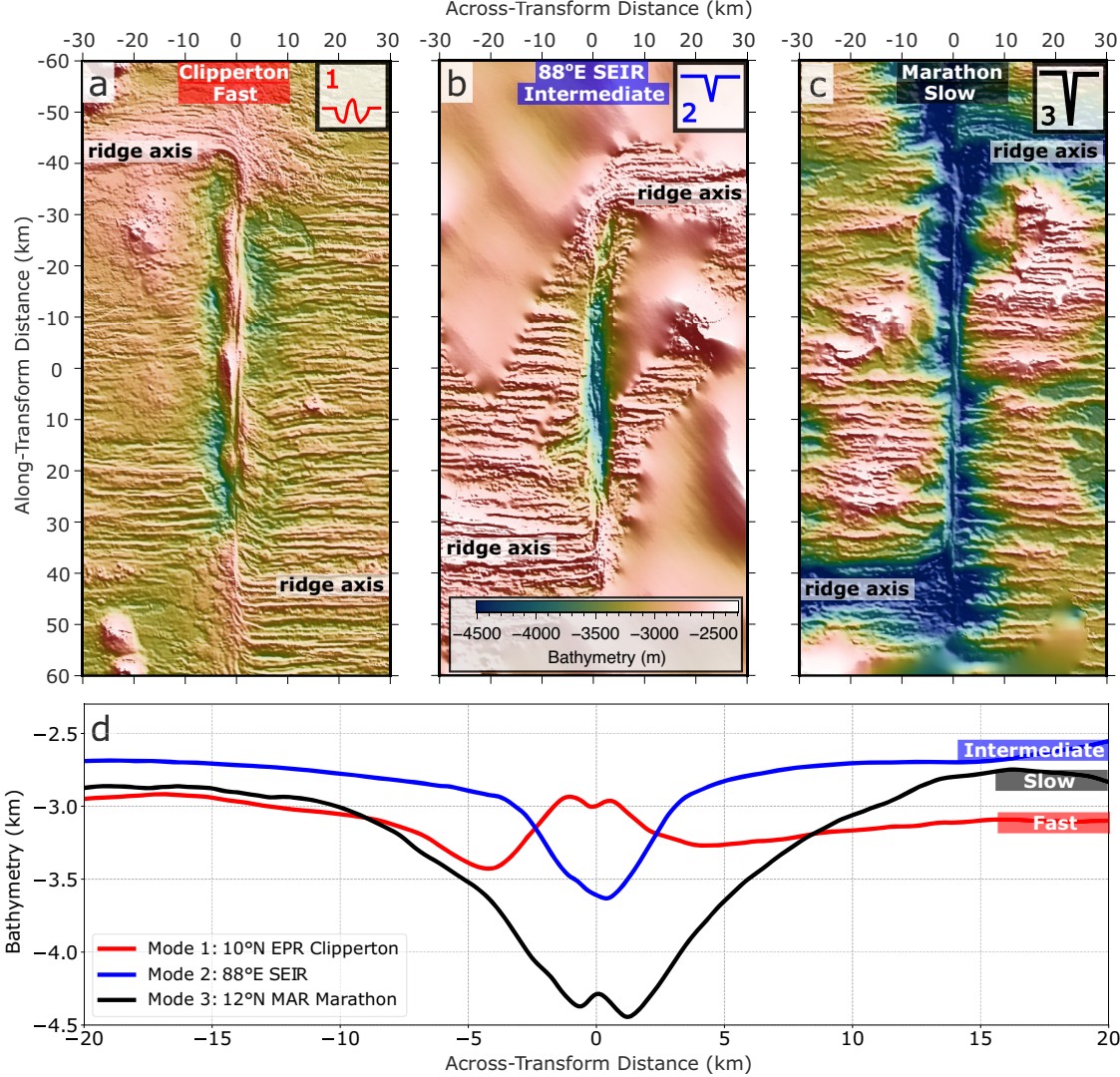

**Fig. 1 | Characteristic bathymetry[49] and transform-perpendicular averaged bathymetric profiles for the three major modes of transform fault morphologies. a** Mode 1, shallow ridge-like topography at 10°N East Pacific Rise (EPR), Clipperton Transform Fault with a full spreading rate of 103.4 mm/yr[3]; **b** Mode 2 intermediate valley at 88°E Southeast Indian Ridge (SEIR) with a full spreading rate of 65.4 mm/yr[3]; and **c** Mode 3 deep valley at 12°N Mid-Atlantic Ridge (MAR), Marathon Transform Fault with a full spreading rate of 24.5 mm/yr[3]. **d** Averaged across-transform topography for each mode shown in **a-c**. The bathymetry maps are rotated so that all transform faults have the same up-down orientation.

between fracture zones and transform valleys. More recently, Grevemeyer et al.[3] proposed that the deepening of transform valleys is due to an age-offset dependent tectonic thinning, which is caused by the increasing obliquity of the transform plate boundary at greater depths. However, this model does not simulate transform topography and uses surface kinematic boundary condition[13,14], which are more appropriate for investigating viscous asthenospheric flow, but not so ideal for addressing strain partitioning in the shallow brittle lithosphere. Relocated seismicity at the fast spreading Gofar transform fault[15,16] and slow spreading Chain transform fault[17] do not show evidence of increasing obliquity of the plate boundary at greater depth, suggesting that the predicted obliquity occurs beneath the brittle–ductile transition.

One important property of mid-ocean ridges that is known to vary with spreading rate is magma supply[18]. Seismic, gravity, and bathymetric observations at the slow-spreading Mid-Atlantic ridge show evidence for thinner crust along transform faults, indicating lower magma supply[19–22]. These observations suggest reduced extension via magmatic intrusions (dikes) and enhanced extension via tectonic faulting near the adjoining ends of the ridge segments, as compared to the segment centers[5,23–25]. In contrast, gravity data at fast-spreading mid-ocean ridges have been used to infer thicker crust along transform faults[20], indicating enhanced magmatism. Because spreading rate-dependent magma supply[18] is well known to affect ridge axis topography through its control on lithospheric thickness and fault style[5,25–30], variations in magma supply may also influence transform fault topography. Consistent with this idea, Grevemeyer et al.[3] showed bathymetric evidence for magmatism extending across the transform fault domain, and linked this to the observed shallowing of fracture zones relative to their adjacent transform valleys[3].

We construct three-dimensional (3-D) numerical models using the finite difference code LaMEM[31] to investigate the origin of oceanic transform topography and its relationship to magma supply (see "Methods" for details). The models simulate a ridge-transform-ridge spreading system with a 6-km thick lithospheric plate and elasto-viscoplastic rheology (Fig. 2). Seafloor spreading is imposed kinematically by pulling on two opposing sides of the model domain, each at a rate $V_x$. Two fixed ridge segments are offset by 32 km, between which a transform fault forms spontaneously as a result of plastic deformation that follows the Drucker–Prager yield criterion. Along the ridge

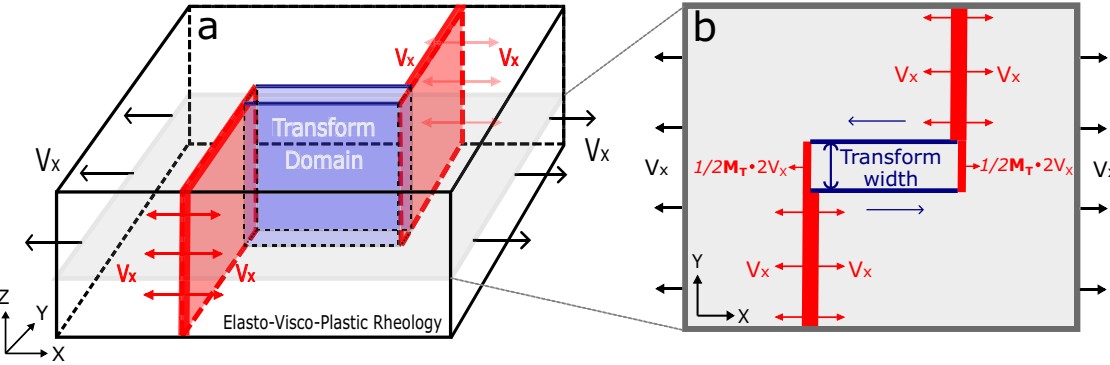

**Fig. 2 | Model setup. a** 3-D ($64 \times 32 \times 8$ km³) model domain with elasto-visco-plastic rheology. Model is composed of cubic elements with 0.5 km edge length. Left and right boundaries are pulled with a half spreading rate of $Vx = 2$ cm/yr. Top boundary is open to flow in and out of a 2- km layer of "sticky air"[45,46]. The other boundaries are free of shear traction with no normal in-and-out flow. **b** Map view of the cross-section shaded in **a**. Two ridge segments open, via diking, at the same rate of plate separation (i.e., $M = 1$) outside of transform domain. Within the transform domain the end of each ridge segment opens at a rate $M_T \cdot Vx$. Transform domain width is set to be either 1 or 2 km. The transform fault strength is governed by Drucker–Prager yield criterion: $\tau_Y = \sin(\varphi)P + \cos(\varphi)C$, where the friction angle $\varphi = 30°$, $P$ is lithostatic pressure and $C$ is cohesion[31].

segments, material divergence is imposed to simulate dike intrusion, which accounts for a fraction, $M$, of the full spreading rate $2 \cdot Vx$[5,23,32]. For the portion of the ridge segment outside of the transform domain, $M = 1$ so that seafloor spreading is fully accommodated by dike intrusions and there is no extensional faulting. This simplification allows us to focus on topography arising from deformation along the transform fault, which is not complicated by abyssal-hill-forming ridge-parallel normal faults[25–27,29,32]. The effects of variable plate separation rate, seafloor depth, transform fault length, and lithospheric thickness are described in the Discussion and Supplementary text.

Inspired by evidence for spreading rate dependent magma supply within oceanic transform faults (e.g., refs. 3,19,20) and high-resolution geological observations from deep-towed photographs of constructional volcanic ridges that traverse across the entire width of the fracture zone at the ridge-transform intersections of the Kane transform[33], we implement dike intrusions along the extensions of the ridge segments within the transform domain. Here, the fraction of seafloor spreading accommodated by diking is denoted as $M_T$, in distinction to $M$, which pertains only to the ridge axes outside of the transform domain (Fig. 2b). The fractional rate of extension due to diking at each end of the transform fault is $M_T/2$, such that $M_T$ represents the combined opening rate of the two transform dike zones. So, when $M_T = 1$, the transform zone dike accretion rate for each segment corresponds to half of the plate separation rate. The transform domain width is set to be either 1 km (for fast-spreading cases) or 2 km (for intermediate to slow-spreading cases), which is roughly consistent with a recent bathymetric analyses[34] that yielded a global median transform width of 2.5 km, but a slightly smaller median width of 1.8 km for systems with a full spreading rate > 8 cm/yr.

Using this model setup, we investigate two primary controls on transform morphology, namely, variations in (1) $M_T$, and (2) fault shear strength, controlled by varying cohesion $C$ (Eq. (2) in "Methods"). We measure model topography on evenly spaced along- and across-fault profiles, from which we calculate the average along- and across-transform model topography (Supplementary Fig. 2d–f for average along- and Supplementary Fig. 2g–i for average across-transform topography for the three example cases shown in Fig. 3). Transform valley and fracture zone depths (Figs. 3d and 4a) are measured from the mean of the along-transform profiles (similar to the analysis of natural systems[3]), once the transform topography has reached steady state or evolve slowly, typically in ≤ 10 million years of model time.

## Results and discussion
### Predicted modes of transform topography
The models show that transform bathymetry deepens systematically with decreasing rates of magmatic accretion ($M_T$) in the transform domain (Fig. 3d). From relatively high to low $M_T$, the models can be categorized into three main modes of transform topography, namely, topographic highs along the transform that are shallower than the adjacent plates (Mode 1), shallow transform valleys (Mode 2), and deep transform valleys (Mode 3). These three modes reproduce the global trend of transform-valley depth vs. spreading rate, and are consistent with the observed range of transform-valley depths (Fig. 3d, e). The models also generate fracture zones that are systematically shallower than transform valleys, as observed globally (Fig. 4).

Mode 1 topography forms when each transform dike intrusion zone accommodates slightly more than half of the seafloor spreading rate ($M_T > 1.02$). This mode shows hundred-meter-high fault-parallel topographic ridges, sometimes separated by small depressions centered along the transform zone or near the ridge-transform intersections (Fig. 3a and Supplementary Fig. 2a, d, g). Higher values of $M_T$ lead to slightly taller positive topographies (Fig. 3d Mode 1). Mode 1 is consistent with the shallow topographic highs observed at some transform faults at fast-spreading ridges (Fig. 1a and Supplementary Fig. 1a, b).

Mode 2 topography occurs when the combined magmatic extension rate in the transform zone approximately matches the seafloor spreading rate ($M_T = 0.98$–$1.02$, Fig. 3d). In these cases, there is little transform-parallel tectonic tension caused by far-field plate separation, as the extension is fully accommodated by magma intrusion. Nonetheless, this mode predicts intermediate depth (100–1000 m) transform valleys (Fig. 3b and Supplementary Fig. 2b, e, h) that deepen with stronger transforms (Fig. 3d Mode 2 and Supplementary Fig. 3). Mode 2 is consistent with intermediate depth transform valleys observed at fast to intermediate spreading ridges where the corresponding fracture zones typically show little relief (Fig. 1b and Supplementary Fig. 1c, d).

Finally, Mode 3 topography occurs when the combined magmatic extension in the transform domain does not fully accommodate the full plate separation ($M_T < 0.98$). This leads to increasing tectonic stretching with decreasing $M_T$ (Fig. 3c & d Mode 3 and Supplementary Fig. 2c, f, i). Mode 3 therefore produces deep valleys (1–4 km), that deepen with decreasing $M_T$ (Fig. 3d). Compared to Mode 2, transform valley depth in Mode 3 models is less sensitive to the transform strength (Supplementary Fig. 3). Further, in contrast to the slightly elevated fracture zone topography in Mode 1 and 2 models, the

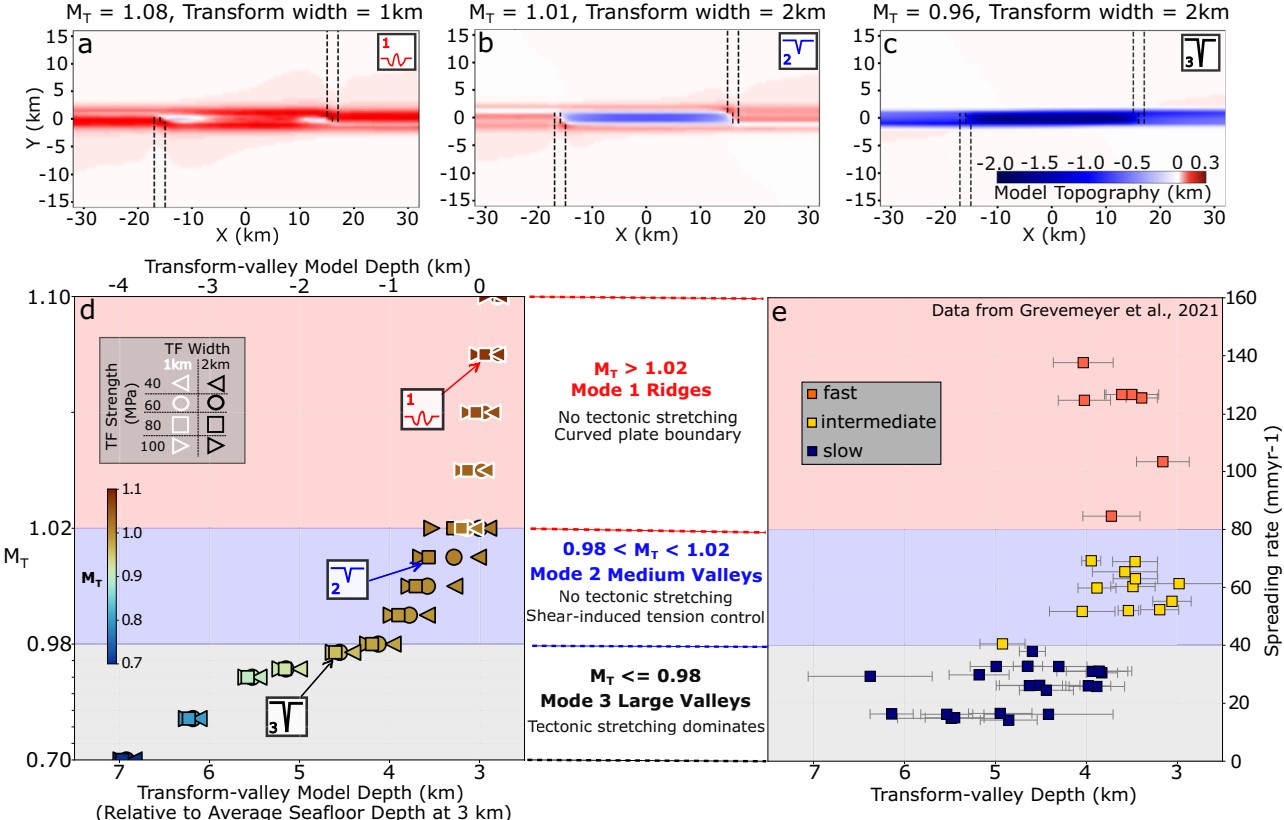

**Fig. 3 | Transform fault morphology as a function of the fraction ($M_T$) of seafloor spreading accommodated by magmatism in the transform domain.** Model topography for **a** a small transform ridge, **b** a shallow transform valley (<1 km relief), and **c** a deep valley (>1 km of relief). Black dashed lines mark the edges of the magma intrusion zones. Note that the positive part of the color scale is stretched to highlight the low amplitude ridge-like topography. Please see Supplementary Fig. 2 for model topography with linear color scale and topographic profiles. **d** Model transform-valley depth as a function of $M_T$ (symbol infill colors using "roma_r" from scientific color maps[50,51]). Symbols denote transform fault cohesion (symbol shape) and imposed transform width (symbol outline color). Transform-valley model depth (top axis) is measured relative to the initial model surface depth (0 km), which is equivalent to 3 km depth on the bottom axis assuming an average seafloor

depth near a ridge axis of 3 km[35-37]. Negative values along the top axis implies subsidence relative to the initial model surface at 0 km, whereas larger values in the bottom axis means deeper seafloor for comparing to data (see Discussion and Supplementary Information for details). Note that variability due to other parameters is tested and is approximately contained within the spread of points for a given value of $M_T$ (see Supplementary Information for details). **e** Global observations of transform valley depth (in km below sea level) as a function of spreading rate[3]. In (d), the vertical axis height of $M_T$ from 0.70 to 0.98 (Mode 3) is scaled by 1/7 (indicated by tick marks with a constant interval of 0.4) so as to match the height of the vertical axis of spreading rate from 0–40 mm/yr in (**e**). This scaling follows the relationship derived by[18,29], which implies that the rate of magmatic extension, M, is the lowest and most variable at the slower spreading ridges.

fracture zones in Mode 3 models are expressed as valleys, but remain ~1 km shallower than the transform valleys (Fig. 4a). Mode 3 is consistent with the deep transform and fracture zone valleys found at slow-spreading rates (Fig. 1c and Supplementary Fig. 1e, f).

### Mechanisms that build transform topography

The model results above indicate a first-order control of intra-transform magmatism on transform fault and fracture zone topography (Figs. 3 and 4). When there is excess magmatism relative to the rate of far-field stretching, the stresses within the transform domain are generally compressive (Mode 1). This causes subtle and time-dependent curvature of the transform fault, which generates local compression resulting in topographic highs (Fig. 5a, b). When magma supply almost perfectly accommodates the far-field tectonic stretching, shear along the fault leads to tension across the transform and the formation of a shallow transform valley (Mode 2). In this case, stronger faults promote deeper valleys (Figs. 3d, Fig. 5c–f and Supplementary Fig. 3) and little subsidence occurs along the adjacent fracture zones (Supplementary Fig. 2e). This across-transform shear-induced tension (Fig. 5c–f) arises first hypothesized by analogy with a rubber band that is pinned on each side of a transform and is stretched due to the strike-slip motion of the fault[12]. The depth of the transform valley scales with this tensile stress, which increases with the shear strength

of the transform fault (Supplementary Fig. 3 ($M_T$ ~ 1) and Fig. 5c, e). Finally, when magma supply within the transform domain is insufficient to accommodate seafloor spreading, tectonic stretching creates deep transform valleys and subsided fracture zones (Fig. 5g, h). In this mode, the transform fault is a hybrid extensional and strike-slip plate boundary.

For most cases except Mode 1 models with the highest $M_T$, the transform valleys are consistently deeper than their adjacent fracture zones (Fig. 4a). The main reason for this is that shear-driven tension only affects the active transform, not the fracture zones. Another contributing factor for the depth difference is that the shear resistance of transform fault causes incomplete lateral opening of the transform dike, which in turn promotes the intruded material to migrate upward (Supplementary Fig. 4). The result is to produce shallower topography in the fracture zone as the new material is accreted and advected away from the active transform fault. Note that while we do not model lava eruptions, this mechanism supports the conceptual idea that magma accretion in the transform domain is important in leading to shallower fracture zones[3].

Oceanic transform faults are observed to form over a range of spreading rates, average seafloor depths, fault lengths, and lithospheric thicknesses. To validate the first-order role of magma intrusion within the transform domain ($M_T$) on transform morphology, and to

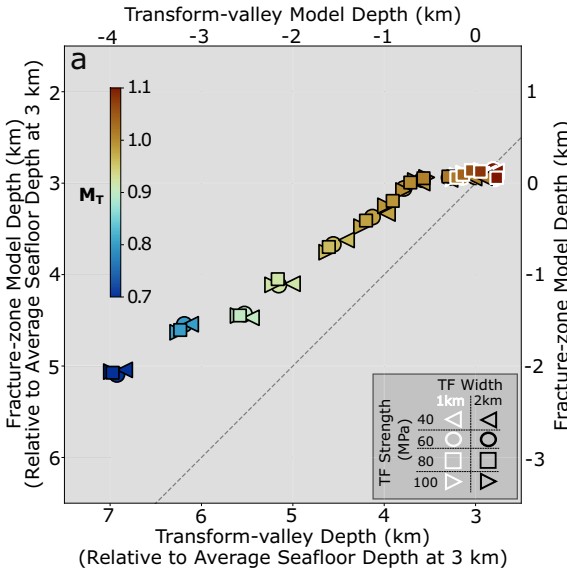

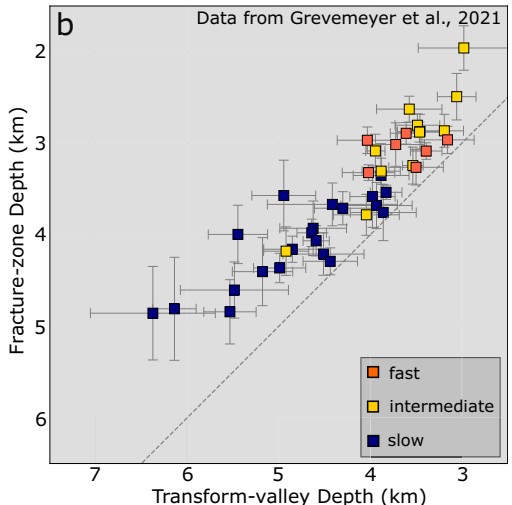

**Fig. 4 | Depth difference between transform faults and adjacent fracture zones.** **a** Model and **b** observed fracture-zone depth versus transform-valley depth (see Supplementary Fig. 2 for measurement details). Model and observed fracture zones are typically shallower than the transform valleys as seen by the vertical shift of the symbols relative to the dashed line, which marks the 1-to-1 ratio. Colors denote $M_T$, symbol shape and outline color identify transform fault strength and width, respectively. Top and bottom horizontal axes are the same as in Fig. 3d. **b** Global observations of transform valley and fracture zone depths (in km beneath sea level) grouped by spreading rate (fast>8 cm/yr, intermediate 4–8 cm/yr and slow <4 cm/yr).

assess the broad applicability of our models, we further investigated the sensitivity of our model predictions to the aforementioned parameters. Of particular importance is that differences in water overburden have negligible effect on model topography (Supplementary Fig. 5 and Table 2). This allows us to infer robust relationships when comparing our model results to observations, even though we assume a constant average seafloor depth of 3 km near mid-ocean ridges (double axes in Figs. 3d and Fig. 4a)[35–37]. Variations in model results due to different half spreading rates within the global range (1, 2, or 5 cm/yr), as well as a factor of ~2 difference in lithospheric thickness (4, 6 or 8 km) or transform fault length (32 or 62 km) are negligible for Mode 1 & 2, but are larger (a few hundred meters) for Mode 3 (Supplementary Fig. 6). These predicted differences may partially explain the larger scatter seen in the observations at slower spreading rates (Figs. 3e and 4b). Meanwhile, we hypothesize that the greater variability in magma supply at slower spreading rates[18] may also be partially responsible for the larger scatter in the natural data. Overall, we find that over this range of parameter space, predicted transform and fracture zone depths cluster closely along the overall trend of the reference model results (Figs. 3 and 4). We therefore conclude that our model results point to $M_T$ as being a globally prominent control on the broad-scale averaged topography of oceanic transform faults.

In addition to these first-order variations in transform topography associated with magmatism, variations in local (e.g., mantle temperature, composition[38]) and/or far-field (e.g., kinematic plate motion[9,10,39]) factors have been proposed to influence transform fault tectonics and generate prominent features such as the median or transverse ridges seen at many oceanic transform faults and fracture zones (ref. 2 for a comprehensive review). For example, median ridges are observed at all spreading rates, including the fast-spreading Clipperton transform (Fig. 1a), the intermediate spreading Chile Ridge transform at 39°S (Supplementary Fig 1c) and Vlamingh transform (Supplementary Fig 1d), as well as the slow-spreading Marathon (Fig. 1c) and Chain transforms (Supplementary Fig 1f)[11]. However, the averaged broad-scale transform topography is not significantly affected by the appearance of median ridges and systematically shallows with faster-spreading rates (Fig. 1d and Supplementary Fig. 1g). The formation of transverse ridges is thought to be

associated with plate motion changes inducing across-transform extension, a process that is not included in our models. However, most transform systems with transverse ridges (e.g., Vema, Chain, and Kane transforms) are still well-classified as Mode 3 systems, with mean across-transform topography characterized by a deep (relief >1 km) valley. Finally, a recent systematic survey of transform fault topography argued that age-offset is a more important control on transform depth than spreading rate[34]. However, because there is a correlation between spreading rate and age offset[4], future work will be needed to distinguish between these two effects.

Overall, our models provide a mechanical basis for a first-order connection between magma supply and the broad-scale topography of global oceanic transform faults and their adjacent fracture zones. This model makes several testable predictions that motivate future investigations. First, local seismicity, detailed seafloor geodesy, high-resolution crustal magnetization and geologic sampling may provide evidence for active magmatic dike intrusions within the transform domain. The predicted stress fields from our models (Fig. 5c, e) are consistent with anomalous focal mechanisms observed at some oceanic transform fault (e.g., thrust mechanism at transform side inside corners; oblique normal faulting adjacent to the transform valleys[40,41]). Second, the dike near the ridge-transform intersection opens asymmetrically, with more material intruded toward the fracture zone side than the transform side (Supplementary Fig. 4). In nature, this could generate thicker crust on the fracture zone side, which is consistent with recent observations showing systematic higher residual mantle Bouguer anomalies indicating thinner crust at transform faults and inside corner regions as compared to the corresponding fracture zones and outside corner regions[42]. Third, our models indicate that stronger faults promote deeper valleys (Fig. 5 c–f, and Supplementary Fig. 3). Because lower earthquake b-values are associated with higher differential stress on a fault[43], this implies that deeper transform valleys should correlate with lower b-values. Recent data from Chain transform fault shows this exact correlation[44], which could be further tested in other systems. Finally, future seafloor geodetic studies might be able to resolve the spreading rate dependence of magmatic accommodated plate extension ($M_T$) within the transform domain, as well as shear-induce extension across the transform.

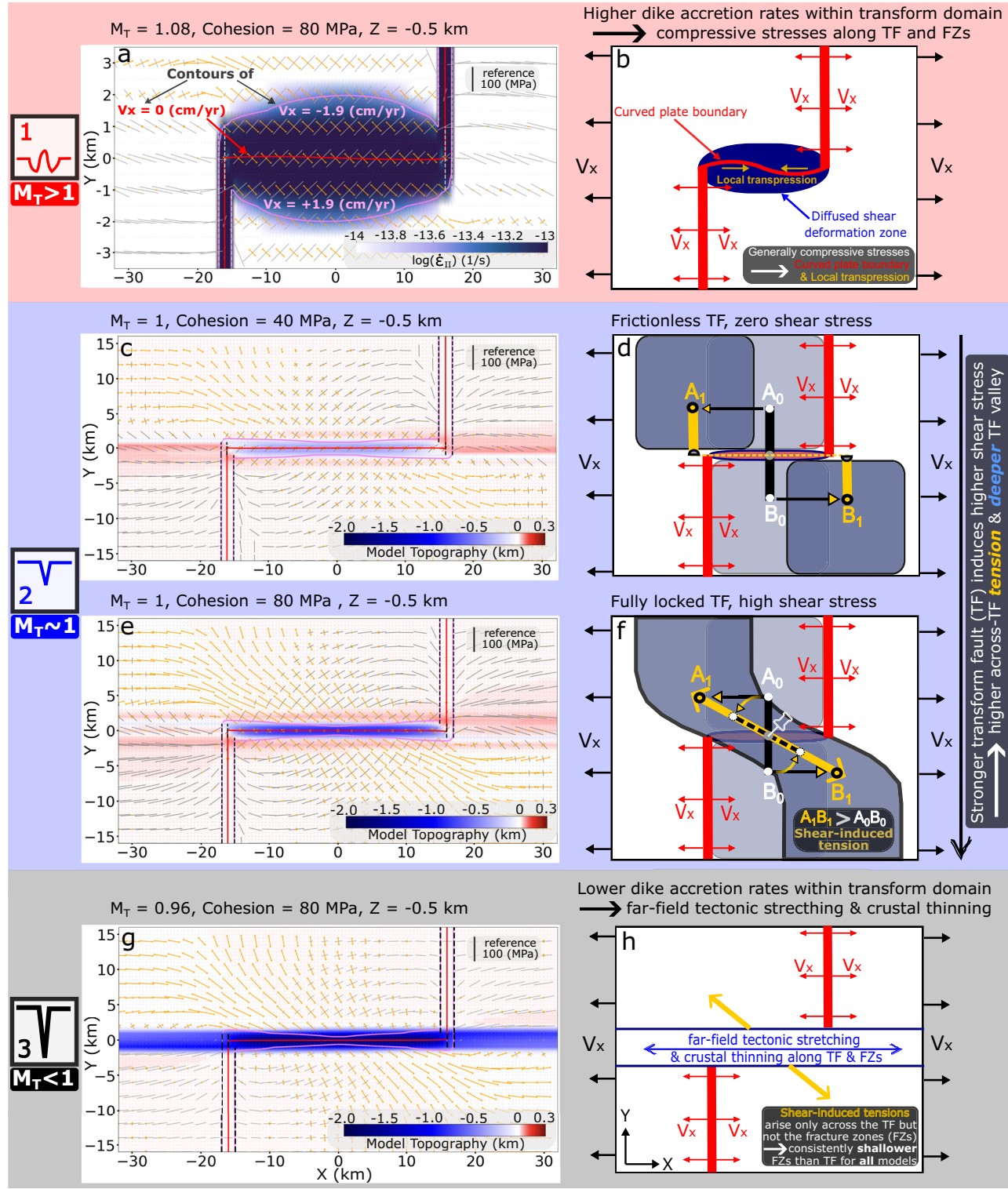

## Methods

We use the open-source numerical code LaMEM (Lithosphere and Mantle Evolution Model)[31]; https://github.com/UniMainzGeo/LaMEM) for the three-dimensional geodynamic simulations. LaMEM employs a finite difference discretization scheme on a fully staggered grid, combined with a marker-in-cell approach to solve the mass and momentum conservation equations using the multigrid numerical method.

The oceanic lithosphere is simulated in a Cartesian model domain of 64 km in $X$-axis (along-transform) direction, 32 km in $Y$-axis (across-transform) direction and 8 km in the vertical $Z$-axis direction (Fig. 2). After systematically testing the effects of grid size on model results, we assume regular cubic mesh with a grid size of 0.5 km. This grid size resolves broad-scale topography of a model that simulates the transform domain as a single shear band along a transform fault. In the vertical $Z$ direction, the base models are composed of 6 km of oceanic lithosphere underlying 2 km of "sticky-air". The "sticky-air" layer is assumed to has a viscosity of $10^{17}Pa \cdot s$ and density of $1 \, \text{kg/m}^3$. The boundary between the "sticky-air" and lithosphere forms an internal free surface[45,46] for tracking the development of topography. Left and

**Fig. 5 | Schematic diagrams illustrating the different mechanisms that control each mode of transform topography.** Magnitude (length of arrows/bars with the reference black bar of 100 MPa) and direction of compressional (gray bars) and tensional (orange arrows) principal stresses as well as velocity contours of Vx = 0 cm/yr (red line) and Vx = +/−1.9 cm/yr (purple lines) are plotted in panels (**a, c, e, g**). For Mode 1, **a** shows a mapview of the square root of the second invariant of the strain rate tensor at 0.5 km depth. Length in Y is exaggerated by a factor of around five to better show the subtly curving plate boundary (red line). Contours of Vx = +/−1.9 cm/yr encompass the diffused shear deformation zone that is under generally compressive stress. The curving plate boundary coincides with local transpression that leads to low-relief uplifted topography as shown schematically in (**b**). For Mode 2, **c** and **e** show principal stresses and velocity contours overlying the model topography. When the fault strength increases with cohesion, the principal tensional stresses (orange arrows) near the transform fault increases from similar to (shown in **c**) to larger than the principal compressional stresses (gray bars) shown

in (**d**). Schematic end member scenarios to illustrate the underlying mechanics: **d** when the transform fault is frictionless, there is no shear stress induced by two plates sliding past each other from time 0 to time 1. The black element $A_0B_0$ is cut in halves which are then translated with the moving plates to the yellow $A_1B_1$. Total length of the element remains the same and so no tensional strain arises across the transform. **f** By contrast, when the transform fault is infinitely strong and allow no slip on the fault, the original black element $A_0B_0$ at time 0 will be elongated at time 1 to the yellow $A_1B_1$ and experience shear-induced tension across the transform. The dashed black line shows the original length of $A_0B_0$ if without tension. For Mode 3, **g** model topography showing transform and fracture zone deepening due to lithospheric thinning from far-field tectonic stretching. **h** For all three modes, the shear-induced tension arises only across the shearing transform fault but not the fracture zones, which partially leads to consistent deeper transform fault than its adjacent fracture zones.

right boundaries are pulled with a half spreading rate of $Vx = 2$ cm/yr. Top boundary is open to in-and-out flow. Other boundaries are shear traction-free and allow no normal in-and-out flow.

As described in ref. [31], the rheology of the lithosphere is assumed to be elasto-visco-plastic and the total deviatoric strain rate is calculated as:

$$\dot{\varepsilon}_{ij} = \dot{\varepsilon}_{ij}^{el} + \dot{\varepsilon}_{ij}^{vs} + \dot{\varepsilon}_{ij}^{pl} = \frac{\tau_{ij}^{J}}{2G} + \dot{\varepsilon}_{II}^{vs}\frac{\tau_{ij}}{\tau_{II}} + \dot{\varepsilon}_{II}^{pl}\frac{\tau_{ij}}{\tau_{II}} \qquad (1)$$

where $\dot{\varepsilon}_{ij} = \frac{1}{2}\left(\frac{\partial v_i}{\partial x_j} + \frac{\partial v_j}{\partial x_i}\right) - \frac{1}{3}\frac{\partial v_k}{\partial x_k}\delta_{ij}$ is the deviatoric strain rate tensor, in which $x_i$ ($i = x, y, z$) denotes Cartesian coordinate in $i$ direction, $v_i$ is velocity in $i$ direction, $\delta_{ij}$ is the Kronecker delta, $\tau_{ij} = \sigma_{ij} + P\delta_{ij}$ is the Cauchy stress deviator tensor ($\sigma_{ij}$ is the Cauchy stress tensor, P is lithostatic pressure). The $\dot{\varepsilon}_{ij}^{el}$, $\dot{\varepsilon}_{ij}^{vs}$ and $\dot{\varepsilon}_{ij}^{pl}$ are the elastic, viscous and plastic components, respectively. $\tau_{ij}^{J} = \frac{\partial \tau_{ij}}{\partial t} + \tau_{ik}\omega_{kj} - \omega_{ik}\tau_{kj}$ is the Jaumann objective stress rate, and $\omega_{ij} = \frac{1}{2}\left(\frac{\partial v_i}{\partial x_j} - \frac{\partial v_j}{\partial x_i}\right)$ is the spin tensor, $G = 40$ GPa is the elastic shear modulus, and the subscript "$II$" denotes the square root of the second invariant of the corresponding tensor.

The magnitude of the plastic strain rate ($\dot{\varepsilon}_{II}^{pl}$) is determined by enforcing the Drucker–Prager yield criterion:

$$\tau_{II} \leq \tau_Y = \sin(\varphi)P + \cos(\varphi)C \qquad (2)$$

where $\tau_Y$ is the brittle yield strength of the oceanic lithosphere, in terms of the second invariant of the deviatoric stress tensor, $\varphi$ is the friction angle of 30°, $P$ is depth-dependent lithostatic pressure and $C$ is cohesion.

A marker and cell method is used to track material properties and material advection is implemented in a Eulerian kinematical framework. During advection, the elastic stress from the previous time step ($\tau_{ij}^{n}$) is corrected on the markers to account for the rigid-body rotation, and then interpolated on the edge and cell control volumes using a distance-based averaging to obtain the effective strain rates:

$$\dot{\varepsilon}_{ij}^{*} = \dot{\varepsilon}_{ij} + \frac{\tau_{ij}^{*}}{2G\Delta t} \qquad (3)$$

where $\tau_{ij}^{*} = \tau_{ij}^{n} + \Delta t(\omega_{ik}\tau_{kj}^{n} - \tau_{ik}^{n}\omega_{kj})$ and $\Delta t$ is the model time step.

The effective viscosity ($\eta^{*}$) and the updated deviatoric stresses ($\tau_{ij}$) are computed from the effective strain rates using the standard

quasi-viscous expression:

$$\tau_{ij} = 2\eta^{*}\dot{\varepsilon}_{ij}^{*}, \eta^{*} = \min\left[\left(\frac{1}{G\Delta t} + \frac{1}{\eta_p}\right)^{-1}, \frac{\tau_Y}{2\dot{\varepsilon}_{II}^{*}}\right] \qquad (4)$$

where $\eta_p = 10^{24} Pa \cdot s$ is the assumed viscosity for the lithospheric plate not at yield. The model setup of an ideal layer with constant thickness and viscosity as a way to isolate second-order effects of asthenosphere drag has been previously used for investigating factors that control normal faulting[47]. This helps reducing the complex competing effects that may complicate the model systematics.

Two mid-ocean ridge segments are offset by 32 km, between which a transform fault forms spontaneously as a result of plastic deformation following the Drucker-Prager yield criterion. The dike intrusion along each mid-ocean ridge segment is implemented as a zone of magmatic intrusion that accounts for a fraction, $M$, of the full plate separation rate $2 \cdot Vx$[5,23,32]. For the ridge segments outside of the transform domain, we assume $M = 1$ such that the far-field tectonic extension is fully accommodated by dike intrusions and the topographic structure is not complicated by abyssal-hill-forming normal faults, which have been extensively investigated previously[26,27,29].

Dike intrusion is implemented within the transform domain based on observations that indicate variable magmatism along oceanic transform faults and fracture zones[3,19,20,42]. The fraction of magmatic intrusion in the transform domain is denoted as $M_T$ to distinguish it from the $M$ value ascribed to the ridge axis outside of the transform domain. At each of the transform magmatic zones, the rate of magmatic extension is $1/2M_T \cdot 2Vx$, and so $M_T \cdot 2Vx$ represents the integrated opening rate within the transform domain, which may have a different relationship with spreading rate than $M$ does for ridge segments outside of the transform domain[18]. The transform domain width is set to be 1 km for Mode 1 cases or 2 km for Mode 2 & Mode 3 cases to reflect the global trend of increasing transform width with slower spreading rate[4,34]. This is roughly consistent with a recent analysis[34] that yielded a global median transform fault width of 2.5 km, but a slightly smaller median width of 1.8 km for systems with a full spreading rate > 8 cm/yr. This approach allows us to simulate the observed increase in transform magma supply with the spreading rate[20].

Using this model setup, we investigate two primary controls on transform morphology: (1) $M_T$, and (2) transform fault shear strength. We measure model topography on evenly spaced along- and across-transform sampling profiles, from which we calculate the average along- and across- transform model topography (Supplementary Fig. 2d - f for mean along- and Supplementary Fig. 2g–i for mean across- transform topography for the three example cases shown in Fig. 3). Transform valley and fracture zone model depths (shown in Figs. 3d and 4a) are measured from the mean along-transform profiles[3], once the across-

transform topography has reached steady state or is evolving slowly, typically within 10 million years of model time.

## Data availability
All data are available in the main text or as Supplementary Figs. and Tables. All source data are provided at https://doi.org/10.5281/zenodo.10552862. Source data for Supplementary Table 1 and 2 are also provided as an excel file "TableS1_S2.xls".

## Code availability
The open-source code LaMEM[31] used for the numerical models in this work is available at https://github.com/UniMainzGeo/LaMEM. All newly coded features for transform domain dike intrusion, input files for the base models and Python plotting scripts can be found at https://doi.org/10.5281/zenodo.10552862.

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

## Acknowledgements

Bathymetry maps in Fig. 1 are plotted with GMT[48]. The spreading velocity of the two fast spreading cases shown in Supplementary Fig. 1a, b is based on the Plate Motion Calculator service provided by the GAGE Facility, operated by EarthScope Consortium, with support from the National Science Foundation, the National Aeronautics and Space Administration, and the U.S. Geological Survey under NSF Cooperative Agreement EAR-1724794. This work benefited from discussions with Eunseo Choi, Jean-Arthur Olive, W. Roger Buck, Lars Rüpke, Ching-Yao Lai, Alexandre Janin and Jizhen Lin. We are grateful for constructive comments from reviewers Fabio Crameri and Milena Marjanović. This work was supported by National Science Foundation grants NSF-OCE 19–28776 to M.B. and X.T.; NSF-OCE 19–28804 to G.I. and J.C.S.; and European Research Council Consolidator grant ERC CoG MAGMA #771143 to B.K. and A.P.

## Author contributions

Conceptualization: M.B., G.I., X.T., and J.C.S.; Methodology: B.K., A.P., J.C.S., and X.T.; Funding acquisition: M.B. and G.I.; Writing – original draft: X.T., M.B., G.I.; Writing – review & editing: X.T., M.B., G.I., J.C.S., B.K., and A.P.

## Competing interests

The authors declare no competing interests.
