## [Peer Review File · Nature Communications]

REVIEWER COMMENTS

Reviewer #1 (Remarks to the Author):

This is an interesting geomechanical modelling study performed with a state-of-the-art numerical code and sensitivity-tested model. The authors reveal how the amount of shallow magma supply is likely a controlling factor differentiating between the three general types of surface morphologies across (oceanic) transform faults (and adjacent fracture zones). This hypothetical geodynamic connection further provides means for additional observations of these often overlooked plate tectonic boundaries.

This is a well reasoned, clear, and very carefully written manuscript. The methods and supplementary sections are useful. I have no major concerns in terms of its scientific accuracy, except for the visual presentation of the data; fixing this is important, but easy. I find this study suitable for the wider science audience of Nature Communications. ^[L1]_[SEP]^[L1]_[SEP]

Lines 31-37: The introduction of the three main types of transform fault topographies is currently only backed up by references to figure 1, which only shows one example for each type. Could these explanations be backed up by a reference that shows that the three morphologies outlined here are indeed globally representative.

Line 219: When you write: „our model results robustly point to MT as the primary control of oceanic transform morphology“: Does this only apply to young (i.e., otherwise undisturbed) transform faults, that have not previously undergone deformation due to other factors, such as major plate rearrangements or formed in pre-existing plate anomalies. If yes, could this be pointed out once to the readers? ^[L1]_[SEP]

Line 350: „The model results are not sensitive to grid sizes smaller than 1 km.“ – I cannot review this. Could you provide these results (maybe in form of a figure to support this statement? How did you test it, and what?

Figure 1: Would it be helpful to also tag each of the panels with the corresponding spreading rates at the ridge, maybe as „fast“, „intermediate“, and „slow“?

Figure 3a,b,c: I see why the color axis is squeezed (i.e., non-uniform), but remind yourselves that doing so misrepresents your dataset as a whole: positive topography is strongly overrepresented (by weighting it >6 times more!) compared to negative topography. Either put a note/reminder to the reader in the caption, that you do this, or use a proper scale (which I recommend since some – if not most – readers only look at the figures). In any case, best to do what you would do for the topography represented on a 2-D profile with a y-axis: would you squeeze the axis ticks on the positive part of the axis too? – Ah, yes, you did, and you decided to make a note to the reader in lines 142-143. Wonderfully illustrates my point that colorbars are not treated the way they should be: as an axis, because they are.

Figure 1d: This panel is not readable by people with colour vision deficiency (and also misrepresents the M_T values: $M_T=1.0$ and $M_T=0.9$ almost look the same while $M_T=0.8$ and $M_T=0.7$ look entirely different). Why not also use a scientific colour map (e.g., from Crameri, 2018), as you already do in figure 1 via GMT? They are easily importable to any other software too. See fabiocrameri.ch/colourmaps on how to do it.

Cramer, F. (2018). Scientific colour maps. Zenodo. <http://doi.org/10.5281/zenodo.1243862>

Figure 4a: See comment on Figure 3a,b,c.

Extended data figures: Same comments as for other figures. Extended data figure 3 has a missing scale for the background topography colouring, which is extra problematic as the data is likely mapped with a distorted scale.

I enjoyed reading the manuscript. – Thank you for all your effort on this!

Reviewer #2 (Remarks to the Author):

Review for manuscript: Magmatism Controls Global Oceanic Transform Fault Topography by Tian et al.

The authors use 3-D numerical modeling to address the observed range in morphology at the oceanic transform faults. The models show that the topography is controlled predominantly by the rate of magma intrusion in the transform domain. For transforms separating fast-spread segments, ridge-like structures characterize the transform. In contrast, in slow-spreading environments, where magma input is limited, the transform domains are represented by deep valleys. The above-described results remarkably match some of the seafloor observations. Moreover, they also reproduce globally observed seafloor depth change from transform fault to fracture zone domain, as was recently highlighted by Grevenmeyer et al. (2021).

The presented models and interpretation are convincing, and I recommend the manuscript for publication in Nature Communication. The only concern I would like to raise is for the authors to consider adding a paragraph to comment on more complex topographies. The models presented in Figure 1a-c) are nice and straightforward, but we often observe complexities in the transform faults topography. For instance, ridge-like structures in the Chain transform fault separate two slow-spreading segments. I understand that this complexity is challenging to model in 3-D, and the authors cannot address every transform system individually. However, it would be good to acknowledge that complex systems exist and that other processes, although not dominant, may be at play (including plate rotation). For instance, could the presence of the ridge-like structure in a slow-spreading environment (such is the case with the Chain TF) be explained by a phase of enhanced magma supply at the bounding ridge segments? Or could they result from the local reorganization of the segments (as suggested by some recent work), for instance? In addition to this concern, below, I list some minor issues that would be good to address before the publication.

Abstract

Line 13: The authors state: "...these faults form tectonically inactive fracture zones...". Based on the earthquake events recorded along the fracture zones some recent studies argue that the fracture zones may not be as passive as we previously thought (e.g., Lay, 2019). It would be good to modify the statement so it acknowledges the existence of this new view.

Lay, T. (2019). "Chapter 4 - reactivation of oceanic fracture zones in large intraplate earthquakes?," in Transform Plate Boundaries and Fracture Zones, ed. J. C. Duarte (Amsterdam: Elsevier), 89–104. doi: 10.1016/B978-0-12-812064-4.00004-9

Figure 1:

It would be good to explain why only these three examples are shown. While these three examples are nice and simple (i.e., represent end members) for modeling purposes, but it needs to be mentioned that more complex systems exist and may be dominated by different or/and composite mechanisms.

In the caption for all spreading rates please add the references.

Line 46: To avoid confusion of using the word ridge for describing Mode 1 and ridge axis for spreading center, please modify to “a) Mode 1 ridge-like topography at 10°N...”.

Lines 50-51: Instead of the sentence: “Bathymetry data from ref 6.”, please provide the reference number on line 45: “Fig. 1. Characteristic bathymetry and transform-...”.

Line 51: The sentence “Plotted with GMT” could go in the Acknowledgement.

Panels a-c) The label ridge axis is hard to see, please use bold characters and introduce white semi-transparent background. Also, it would be good to indicate the name of the transform fault in each panel.

Section: Prior models for transform fault morphology

Lines 64-65: The authors state: “ Seismic, gravity, and bathymetric observations at the slow-spreading Mid-Atlantic ridge show evidence for thinner crust along transform faults and fracture zones, indicating lower magma supply.”

The compilation of the available seismic data collected at the Mid-Atlantic Ridge, including some recent studies, show that for the fracture zones we do not see crustal thinning, i.e., for most of the fracture zones crustal thickness is > 5 km. Please see Figure 11 in Marjanović et al. (2020).

Please modify the statement to agree with the observations (e.g., you could take out fracture zones).

Marjanović, M., Singh, S. C., Gregory, E. P. M., Grevemeyer, I., Growe, K., Wang, Z., et al. (2020). Seismic crustal structure and morphotectonic features associated with the Chain Fracture Zone and their role in the evolution of the equatorial Atlantic region. *Journal of Geophysical Research: Solid Earth*, 125, e2020JB020275. <https://doi.org/10.1029/2020JB020275>

Section: 3-D transform models incorporating magma intrusion

In the first paragraph you describe the model using 6 km thick crust and 32 km long transform. It would be good if you would inform the reader that models using different parameters were also considered, described in the supplementary material, and discussed later in the text.

Figure 2: Caption – when describing the model, the authors state that the vertical Z-axis is 8 km and refer to Figure 2. However, in the figure caption you indicate that the model dimensions are 64x32x6 km (also, it should be km³). For consistency, please modify the caption.

Reviewer #3 (Remarks to the Author):

This is a well written and presented manuscript using numerical modelling to show how differing amounts of magmatism may control the morphology of transform faults and fracture zones. This morphology can be some of the most variable and extreme anywhere on the ocean floor (perhaps on all of Earth), and so these exciting results presenting a possible mechanism for explaining some of this variability are significant and interesting to a wide variety of Earth scientists. Generally, I think that the manuscript is in a good shape. My main comments are that the narrative is perhaps oversimplified in a few different areas and there are a few oversights that should be addressed. The numerical modelling is outside of my area of expertise.

1. Some of the most extreme morphology with large ridges, often known as transverse ridges, at transform faults and fracture zones has been seen at places such as St Paul, Romanche, Chain, Vema, or at least is mostly seen at slow-spreading ridges. Most commonly, much of this topography has been thought to be due to transpression/tension due to non-orthogonal plate movements (e.g. Maia et al. 2016; Marjanovic et al. 2020). Many of these areas (e.g. Romanche) have been thought to have very low levels of magmatism, though this is now in debate with some evidence for more magmatism than previously thought (e.g. Gregory et al., 2021). The authors should comment on these features at slow spreading ridges as they are different to the Mode 1 ridges produced in the modelling – in these locations is magmatism likely also the primary driver? Even though, for example, the St Paul rocks have been shown to consist of peridotites and mylonites (Maia et al., 2016)? Or, if there are plate tectonic directional controls, do these override the original magmatic controls on morphology? This is an interesting question that should at least be addressed. Along with the tectonic controls, there are also potential effects on topography from the composition of the lithosphere, and the age-offset relationship proposed by Ren et al. (2022) which are not discussed, which is an oversight. Magmatism may be an important or even primary driver, but that doesn't mean these other mechanisms become insignificant or have no effect.

2. Please add a more detailed explanation as to why fracture zones are shallower than transform faults in the model as these mechanisms weren't clear to me, and this is a significant and interesting result that warrants a proper explanation. It is mentioned (line 194) that there is less extension there as it is stronger, but doesn't the already deep transform fault then become a fracture zone? So how does it become more shallow? The <1% asymmetry in dike intrusion is not enough to raise seafloor depth significantly so cannot be the explanation in the model. Please clarify in the manuscript or if you do not know the mechanisms then state this explicitly.

3. How possible or likely is it to have $M_t > 1$? Is there precedent for this in other studies of magmatic vs tectonic spreading at mid-ocean ridges as I assumed the maximum value of M was 1? I am also wondering about the likelihood of having more magmatism in the transform domain than along the main ridge segment ($M_t > M$). Is this relative relationship (M_t vs M) significant – i.e. higher or lower rates of magmatism within the transform domain when compared to the ridge, rather than the M_t value alone? Please add something to justify these values.

4. Generally, this study is modelling transform faults as hybrid extensional-transform plate boundaries, rather than being solely strike-slip, with full spreading occurring in the transform zone. This is only mentioned in the last sentence of the abstract but is a significant feature of the study and should be addressed in the discussion section also.

5. I found Extended data Figure 3 very helpful in understanding the mechanisms for forming the topography in the different modes. It would be better to have this figure in the main text.

6. Relating to the sensitivity tests done on water overburden, these are explained in a misleading way (paragraph starting line 205; line 492; line 561) which suggests that multiple water depths have been tested. My understanding from the text is that only no water and 2 km of water have been tested – thus what has been tested is the uncertainty in the model results for not taking into account the density of water, not changes in seafloor depth. Please change all parts of the manuscript/supplement where this is explained to clarify this, or better, run at least one more test with a different seafloor depth (e.g. 3 km) with water density included.

7. Relating to the other sensitivity tests done – these are good and are a beneficial part of the study to help understand the effects of these different parameters on the TF/FZ morphology. I found the way the results are explained/presented not ideal in the main manuscript and the supplementary text – it would be better to have these variations also plotted as error bars on the model data in figures 3a and 4a please. Then they can be compared by eye to the variations in the real data in 3b and 4b. Also, in sentences starting lines 212 and 215, I wouldn't agree that changes on the order of 'only' hundreds of metres are insignificant or mean the predictions are 'insensitive' (though I realise they are less than this for two of the modes). It would be better and more useful to the reader to present these results/parameters as meaningful in some scenarios and perhaps responsible for some of the other variation seen in observed data, even if secondary to the effect of magmatism. This comment relates both to the section in the main manuscript and how the sensitivity test results are presented and discussed in the supplement. I'm not sure I agree there is much support for the hypothesis on line 223 – some of this scatter could be related to the tested parameters, as well as other things that can affect TF/FZ topography such as non-orthogonal far-field plate tectonic stresses.

8. In general, there are more grammatical errors and the text is less clear in the Supplementary section. Please go over and edit this more carefully.

Other comments by line

Line 53: I find this section explaining that only two mechanisms are invoked a little oversimplified, and ignore the recent studies of Grevemeyer et al. (2021) and Ren et al. (2022), aside from a small comment about the shallowing of fracture zones. These recent papers and their findings should be included in more detail in this section.

Line 57-58: Are there examples/references for this claim for transform ridges?

Line 68-69: Have there been any seismic studies that can confirm this idea of an increase in crustal thickness at fast-spreading transforms? Studies I am aware of (e.g. at Gofar & Quebrada on the EPR, Roland et al (2012)) don't report thicker crust, so please add 'potentially' or similar. Or add some more references for support here.

Line 131 (Figure 3): Why weren't all strength and width combinations tested? E.g. widths of 2 km at $M_t > 1.02$. Also, given that the y axis is variable in (d), please add some small tick marks with a regular interval (e.g. every 0.05), so this variability/non-linearity is clear.

Line 139-141: Sentence beginning 'Negative values' needs rewording, I take it you mean the top axis is difference in depth from the initial surface, and the bottom axis is the absolute depth.

Line 165: Sentence beginning 'Compared to Mode 2'. Please slightly reword this sentence for clarity, e.g. 'Mode 2 valleys depths are more sensitive to transform strength than Mode 3, increasing in depth with increasing transform strength', or another wording of your choice.

Line 232: The reported asymmetry is less than 1% - I don't see how that is significant enough to make a measurable impact on crustal thickness. Please either elaborate on this e.g. could higher percentages of asymmetry be possible for particular reasons, or change/remove this.

Methods/supplementary sections

Line 404: The choice of 1 and 2 km wide transforms is confusing – most transforms in the reference quoted, Ren et al (2022), are wider than 2 km (average 3.7 km) and why only look at those with a spreading rate >8 cm/year? Ren doesn't include fast-spreading transforms – if this is why then please explain in the text. Also, why is a median quoted as a range, shouldn't this be a single value? It would have been interesting to see the effects of wider transforms, as is more common at slow spreading rates.

Line 451: Change to 'time 1 (t1) and time 2 (t2)' so the reader knows what these are in the figure.

Line 461: Sentence starting 'With water' is confusing and the numbers given don't all match those in Extended Data 2, which I'm guessing they should? Please rewrite to clarify and correct the numbers, e.g. 'shallower and deeper' are the wrong way around for Mode 1.

Line 469: Please change 'stroke colour' to 'outline' throughout this caption.

Line 507: Add 'observed' before data – as it needs to be clear you are comparing to the real observed data.

Lines 497 and 520: Please give a value in parentheses for these 'negligible changes' e.g. (<10 m).

Line 526: Would be useful to add an explanation as to why these changes in spreading rate have this effect on the depths.

Line 529 and line 544: Sentences starting 'Also, these variations' and 'The changes are also...'. Please remove these sentences - if the trends are the same, I don't think this is relevant to state.

References:

Gregory, E. P. M., Singh, S. C., Marjanović, M., & Wang, Z. (n.d.). Evidence for thick mafic crust at the slow-slipping Romanche oceanic transform fault. Geology. 2021. <https://doi.org/10.1130/G49097.1>

Grevemeyer, I., Rüpke, L. H., Morgan, J. P., Iyer, K., & Devey, C. W. (2021). Extensional tectonics and two-stage crustal accretion at oceanic transform faults. *Nature*, <https://doi.org/10.1038/s41586-021-03278-9>

Maia, M., Sichel, S., Briaies, A., Brunelli, D., Ligi, M., Ferreira, N., Campos, T., Mougel, B., Brehme, I., Hémond, C., Motoki, A., Moura, D., Scalabrin, C., Pessanha, I., Alves, E., Ayres, A., & Oliveira, P. (2016).

Extreme mantle uplift and exhumation along a transpressive transform fault. *Nature Geoscience*, <https://doi.org/10.1038/ngeo2759>

Marjanović, M., Singh, S. C., Gregory, E. P. M., Grevemeyer, I., Growe, K., Wang, Z., Vaddineni, V., Laurencin, M., Carton, H., Gómez de la Peña, L., & Filbrandt, C. (2020). Seismic Crustal Structure and Morphotectonic Features Associated With the Chain Fracture Zone and Their Role in the Evolution of the Equatorial Atlantic Region. *Journal of Geophysical Research: Solid Earth*, <https://doi.org/10.1029/2020jb020275>

Ren, Y., Geersen, J., & Grevemeyer, I. (2022). Impact of Spreading Rate and Age-Offset on Oceanic Transform Fault Morphology. *Geophysical Research Letters*, <https://doi.org/10.1029/2021GL096170>

Roland, E., Lizarralde, D., McGuire, J. J., & Collins, J. A. (2012). Seismic velocity constraints on the material properties that control earthquake behavior at the Quebrada-Discovery-Gofar transform faults, East Pacific Rise. *Journal of Geophysical Research: Solid Earth* <https://doi.org/10.1029/2012JB009422>

**REVIEWER COMMENTS are in blue**

**Replies are in black**

-----
**We thank all the reviewers for their thoughtful comments.**

One suggestion raised by all the reviewers is that we should more thoroughly discuss the role of
mechanisms other than magmatism on transform fault topography. We have revised the
manuscript following this suggestion. In particular, we try to better emphasize that the study
focuses on the broad-scale and long-term averaged topographies of oceanic transform faults as
well as their general dependence on the transform domain magmatism, which scales with
spreading-rate. Meanwhile, we acknowledge that our models do not explain some of the
smaller-scale and temporally variable complexity seen at individual transform faults.

Specifically, we have made the following major modifications:

- 1. We modify the text with additional references to clarify that we are focusing on the
global trend of broad-scale topography rather than the relatively local prominent
morphological features like median ridges or transverse ridges.
- 2. We added a Supplementary Fig.1 to illustrate that the overall trend of broad-scale
topography with spreading rates shown in main text Fig. 1 is observed across
multiple transforms. Including the 3 examples shown in Fig. 1, we now show 9
examples (3 for each mode).
- 3. We added a paragraph to the Discussion regarding the effects of other factors on
generating various transform fault morphologies and their influence on the broad-
scale topographic variations observed as a function of spreading rate.

We discuss the specific changes we made to address this broad concern as well as a point-by-
point response to the reviewers' individual comments below.

**In addition, following the requirements from the Nature Communications formatting**
**instructions, we made the following changes to the structure of the paper:**

- 1. We shorten the abstract to below 150 words.
- 2. Because Nature Communications allow up to 10 display items, following suggestion from
reviewer 3, we moved the original Extended Data fig. 3 to the main text as the current
Figure 5.
- 3. All other original Extended Data Figures and Tables are now in Supplementary
Information as Supplementary Figures and Tables.

-----
**Reviewer #1 (Remarks to the Author):**

This is an interesting geomechanical modelling study performed with a state-of-the-art
numerical code and sensitivity-tested model. The authors reveal how the amount of shallow
magma supply is likely a controlling factor differentiating between the three general types of
surface morphologies across (oceanic) transform faults (and adjacent fracture zones). This
hypothetical geodynamic connection further provides means for additional observations of these
often overlooked plate tectonic boundaries. This is a well reasoned, clear, and very carefully
written manuscript. The methods and supplementary sections are useful. I have no major
concerns in terms of its scientific accuracy, except for the visual presentation of the data; fixing
this is important, but easy. I find this study suitable for the wider science audience of Nature
Communications.

Lines 31-37: The introduction of the three main types of transform fault topographies is currently
only backed up by references to figure 1, which only shows one example for each type. Could
these explanations be backed up by a reference that shows that the three morphologies
outlined here are indeed globally representative.

Thanks for pointing out this issue, which is also raised by Reviewer 2. To emphasize that these
broad systematic changes in topography are seen across a range of transform systems, we
added a figure (Supplementary Fig. 1) with 6 more examples (2 additional examples for each
type) that show similar results as compared to the Fig. 1 in the main text. We also added the
following four references in a modified first sentence of the Introduction:

“Oceanic transform faults display a wide range of topographic morphologies^{1,2}. Overall, the
broad-scale topographies of transform faults deepen systematically with decreasing spreading
rate^{3,4} (Fig. 1d, Fig. 3e and Supplementary Fig. 1g). ”

1. Wolfson-Schwehr, M. & Boettcher, M. S. Global characteristics of oceanic transform fault
structure and seismicity. in *Transform plate boundaries and fracture zones* 21–59 (Elsevier,
2019).
2. Maia, M. Topographic and morphologic evidences of deformation at oceanic transform
faults: far-field and local-field stresses. in *Transform plate boundaries and fracture zones* 61–87
(Elsevier, 2019).
3. Grevemeyer, I., Rüpke, L. H., Morgan, J. P., Iyer, K. & Devey, C. W. Extensional
tectonics and two-stage crustal accretion at oceanic transform faults. *Nature* **591**, 402–407
(2021).
4. Luo, Y., Lin, J., Zhang, F. & Wei, M. Spreading rate dependence of morphological
characteristics in global oceanic transform faults. *Acta Oceanologica Sinica* **40**, 39–64 (2021).

In addition, after carefully examining the literature, including the references above and by
considering the comments from all reviewers (described above) regarding complex transform
structures caused by other factors, we revised our description of the three modes of
morphologies. Specifically, there was confusion over our originally used term ‘transform ridges’,
which can mean ‘median ridges’ within the transform domain or ‘transverse ridges’ bounding the
transform valley. Thus, we revised our description for the fast-spreading cases to prevent
confusion and focus on the broad-scale average topography:

“At the fastest seafloor spreading rates, the transform zone is relatively shallow, sometimes even
shallower than the adjacent seafloor (Mode 1, Fig. 1a, Supplementary Fig. 1).”

Second, ‘median ridges’ within the transform faults are observed across all spreading
regimes, and their origin is controversial and may be case-dependent. However, as we now
discuss in the added paragraph in the Discussion section, in most circumstances their
appearance does not overwrite the average broad-scale spreading-rate dependent
topography which is the focus of this study.

Lastly, we also added a sentence following descriptions of the 3 modes for clarification:

“Note that while we broadly categorize transform topography into 3 discrete modes,
these variations with spreading rate occur on a continuum. Moreover, other processes
may be at play for generating complex smaller-scale, time-dependent morphologies.”

Line 219: When you write: „our model results robustly point to MT as the primary control of
oceanic transform morphology“: Does this only apply to young (i.e., otherwise undisturbed)

transform faults, that have not previously undergone deformation due to other factors, such as
major plate rearrangements or formed in pre-existing plate anomalies. If yes, could this be
pointed out once to the readers?

Thank you. This point was raised by all reviewers. As described above, although other factors
clearly affect transform fault topography (which we now described further in the added
discussion near the end of the main text), our results focus on explaining the global variation in
broad-scale transform fault topography as a function of magma supply. We have revised the
sentence to the following:

" We therefore conclude that our model results point to M_T as being a globally prominent
control on the broad-scale averaged topography of oceanic transform faults. "

This above sentence is then followed by the newly added paragraph that comprehensively
discuss effects of other factors and their influence on smaller-scale morphologic features.

Line 350: "The model results are not sensitive to grid sizes smaller than 1 km."—I cannot review
this. Could you provide these results (maybe in form of a figure to support this statement? How
did you test it, and what?

Good point. We tested the sensitivity of model results on variable grid sizes and we present
those results below using the same value ranges for the X and Y axes from Fig. 3d and Fig. 4a
in the main text:

Reply figure for grid size sensitivity. The rectangles represent 3 reference models (Fig. 3 a, b, c
from the main text) with cubic grid and edge length of 0.5 km. Stars, circles and crosses show
results from models with cubic grid near the transform fault whose edge length are 1 km, 0.66
125 km and 0.25 km respectively.

Overall, we found that models with grid sizes less than 1 km show consistent results that
support the first-order control of M_T on the broad-scale transform fault and fracture zone
topography. For grid size less than or equal to 0.66 km, the variability of our model results with

respect to grid resolutions is similar to the variability associated with other parameters like
lithospheric thickness, spreading rate and transform length (Supplementary Fig. 6). For mode II,
the case with grid size of 0.25 km predicted multiple anastomosing shear bands, thus explaining
the shallower valley depth. At this resolution the model is beginning to resolved smaller-scale
deformation within the transform zone. Indeed, small-scale deformation features are observed
within natural transform faults, but resolving those would require even finer grid resolution,
which would be too computationally expensive, and is not the topic of this study. A grid
resolution of 0.5 km is sufficient to resolve the large-scale topography as a function of magma
supply.

Meanwhile, we have revised the sentence pointed out by the reviewer to better reflect the
details described above as:

*“After systematically testing the effects of grid size on model results, we assume regular cubic*
*mesh with a grid size of 0.5 km. This grid size resolves broad-scale topography of a model that*
*simulates the transform domain as a single shear band along a transform fault.”*

Figure 1: Would it be helpful to also tag each of the panels with the corresponding spreading
rates at the ridge, maybe as „fast“, „intermediate“, and „slow“?

Added accordingly.

Figure 3a,b,c: I see why the color axis is squeezed (i.e., non-uniform), but remind yourselves
that doing so misrepresents your dataset as a whole: positive topography is strongly
overrepresented (by weighting it >6 times more!) compared to negative topography. Either put a
note/reminder to the reader in the caption, that you do this, or use a proper scale (which I
recommend since some – if not most – readers only look at the figures). In any case, best to do
what you would do for the topography represented on a 2-D profile with a y-axis: would you
squeeze the axis ticks on the positive part of the axis too? – Ah, yes, you did, and you decided
to make a note to the reader in lines 142-143. Wonderfully illustrates my point that colorbars are
not treated the way they should be: as an axis, because they are.

Thank you for pointing out these issues. Here were our original goals when we crafted Fig.3
a,b,c:

- 1. When model topography is zero, we wanted to use white color to emphasize there are
no topographic changes.
- 2. We wanted to keep the same color scales for all model topography plots so as to make it
easy to compare differences in the amplitude of topography between cases.
- 3. We squeezed the positive topography color scale because we wanted to highlight the
positive ridge-like topography, even though we acknowledge that it has one order
magnitude smaller amplitude compared to mode 3 valleys. Indeed, without this
modification the positive topography does not show up well when using a linear -2 to 2
170 km color scale (please see the updated Supplementary Fig. 2 attached below).

To best achieve these goals, while mitigating the possibility that a reader may misinterpret the
data as pointed out by the reviewer, we made the following changes:

- 1. In Supplementary Fig. 2 (previously Extended Data Fig. 1d-i) we show model
topography with linear -2 to 2 km color scale along with topographic profiles in which the
vertical axis that is linearly proportional to the topographic values.
- 2. As suggested by the reviewer, we also mention explicitly in the Main Text Fig. 3 caption
that: *“Note that the positive part of the color scale is stretched to highlight the low*
*amplitude ridge-like topography. Please see Supplementary Fig. 2 for model topography*
*with linear color scale and topographic profiles.”*

Updated Supplementary Fig. 2:

Figure 3d: This panel is not readable by people with colour vision deficiency (and also
misrepresents the M_T values: $M_T=1.0$ and $M_T=0.9$ almost look the same while $M_T=0.8$
and $M_T=0.7$ look entirely different). Why not also use a scientific colour map (e.g., from
Crameri, 2018), as you already do in figure 1 via GMT? They are easily importable to any other
software too. See fabiocrameri.ch/colourmaps on how to do it.
Crameri, F. (2018). Scientific colour maps. Zenodo. <http://doi.org/10.5281/zenodo.1243862>
Following the suggestions, we changed the color map of Figure 3d to “roma_r” from Crameri, F.
(2018) with citations added: “symbol infill colors using ‘roma_r’ from scientific color maps
(Crameri, 2018; Crameri et al., 2020)”.

Figure 4a: See comment on Figure 3a,b,c.

According to suggestion, we changed the color map of Figure 4a to “roma_r” from Crameri, F.
(2018).

Extended data figures: Same comments as for other figures. Extended data figure 3 has a
missing scale for the background topography colouring, which is extra problematic as the data is
likely mapped with a distorted scale.

We have updated the color map following the above suggestions for Supplementary Figures
(previously Extended Data Fig. 2, 4, 5). In the revised manuscript we moved Extended Data
Fig. 3 to the main text (now Fig. 5) and added the missing color scale. Please also see
explanations above for the reason for choosing the color scale.

I enjoyed reading the manuscript. – Thank you for all your effort on this!

-----

Reviewer #2 (Remarks to the Author):

Review for manuscript: Magmatism Controls Global Oceanic Transform Fault Topography by
Tian et al.

The authors use 3-D numerical modeling to address the observed range in morphology at the
oceanic transform faults. The models show that the topography is controlled predominantly by
the rate of magma intrusion in the transform domain. For transforms separating fast-spread
segments, ridge-like structures characterize the transform. In contrast, in slow-spreading
environments, where magma input is limited, the transform domains are represented by deep
valleys. The above-described results remarkably match some of the seafloor observations.
Moreover, they also reproduce globally observed seafloor depth change from transform fault to
fracture zone domain, as was recently highlighted by Grevemeyer et al. (2021).
The presented models and interpretation are convincing, and I recommend the manuscript for
publication in Nature Communication.

-----

The only concern I would like to raise is for the authors to consider adding a paragraph to
comment on more complex topographies. The models presented in Figure 1a-c) are nice and
straightforward, but we often observe complexities in the transform faults topography. For
instance, ridge-like structures in the Chain transform fault separate two slow-spreading
segments. I understand that this complexity is challenging to model in 3-D, and the authors
cannot address every transform system individually. However, it would be good to acknowledge
that complex systems exist and that other processes, although not dominant, may be at play
(including plate rotation). For instance, could the presence of the ridge-like structure in a slow-
spreading environment (such is the case with the Chain TF) be explained by a phase of
enhanced magma supply at the bounding ridge segments? Or could they result from the local
reorganization of the segments (as suggested by some recent work), for instance? In addition to
this concern, below, I list some minor issues that would be good to address before the
publication.

Thanks for pointing out this issue. As described above, we have addressed this issue by:

- 1. Adding a paragraph in Discussion describing the effects of other factors on transform fault
topography.
2. Revising our descriptions of ridge-like structure shown in the Mode 1 cases to clarify our
focus on broad-scale topography.
3. Noting at several locations in the text that other factors may be at play (e.g., in the
Introduction)

**Abstract**
Line 13: The authors state: "...these faults form tectonically inactive fracture zones...". Based on
the earthquake events recorded along the fracture zones some recent studies argue that the
fracture zones may not be as passive as we previously thought (e.g., Lay, 2019). It would be
good to modify the statement so it acknowledges the existence of this new view.
Lay, T. (2019). "Chapter 4 - reactivation of oceanic fracture zones in large intraplate
earthquakes?," in Transform Plate Boundaries and Fracture Zones, ed. J. C. Duarte
(Amsterdam: Elsevier), 89–104. doi: 10.1016/B978-0-12-812064-4.00004-9

Thanks for pointing out this new view. Because Nature Communications has a 150-word limit
for abstract, we now have removed this sentence in the new version.

Figure 1:

It would be good to explain why only these three examples are shown. While these three
examples are nice and simple (i.e., represent end members) for modeling purposes, but it
needs to be mentioned that more complex systems exist and may be dominated by different
or/and composite mechanisms.

Thanks for pointing this issue out which is also raised by Reviewer 1. As described above, we
added Supplementary Figure 1 with 6 additional examples of transform fault topography
including 2 new examples for each mode to support the Fig.1 in the main text.

We also added discussion near the end of the main text regarding other factors that could be
important in generating some of the more complex transform fault morphologies.

In addition, we added 2 sentences in the Introduction to clarify the point raised by the reviewer:
*“Note that while we broadly categorize transform topography into three discrete modes, these*
*variations with spreading rate occur on a continuum. Moreover, other processes may be at play*
*for generating complex small-scale, time-dependent morphologies².”*

In the caption for all spreading rates please add the references.

Added accordingly for both Fig. 1 and the newly added Supplementary Fig. 1.

Line 46: To avoid confusion of using the word ridge for describing Mode 1 and ridge axis for
spreading center, please modify to “a) Mode 1 ridge-like topography at 10°N...”.

Changed accordingly.

Lines 50-51: Instead of the sentence: “Bathymetry data from ref 6.”, please provide the
reference number on line 45: “Fig. 1. Characteristic bathymetry⁶ and transform-...”.

Changed accordingly.

Line 51: The sentence “Plotted with GMT” could go in the Acknowledgement.

Changed accordingly.

Panels a-c) The label ridge axis is hard to see, please use bold characters and introduce white
semi-transparent background. Also, it would be good to indicate the name of the transform fault
in each panel.

Changed accordingly.

Section: Prior models for transform fault morphology

Lines 64-65: The authors state: “ Seismic, gravity, and bathymetric observations at the slow-
spreading Mid-Atlantic ridge show evidence for thinner crust along transform faults and fracture
zones, indicating lower magma supply.”The compilation of the available seismic data collected
at the Mid-Atlantic Ridge, including some recent studies, show that for the fracture zones we do
not see crustal thinning, i.e., for most of the fracture zones crustal thickness is > 5 km. Please
see Figure 11 in Marjanović et al. (2020).Please modify the statement to agree with the
observations (e.g., you could take out fracture zones).

Marjanović, M., Singh, S. C., Gregory, E. P. M., Grevemeyer, I., Growe, K., Wang, Z., et al.
(2020). Seismic crustal structure and morphotectonic features associated with the Chain
Fracture Zone and their role in the evolution of the equatorial Atlantic region. Journal of

Geophysical Research: Solid Earth, 125,
e2020JB020275. [https://doi.org/ 10.1029/2020JB020275](https://doi.org/10.1029/2020JB020275)
Changed accordingly by removing “fracture zones” and adding the suggested reference.

Section: 3-D transform models incorporating magma intrusion

In the first paragraph you describe the model using 6 km thick crust and 32 km long transform. It
would be good if you would inform the reader that models using different parameters were also
considered, described in the supplementary material, and discussed later in the text.

We added the following sentence at the end of the paragraph: “*Note that the effects of variable*
*spreading rate, seafloor depth, transform fault length, and lithospheric thickness are described*
*in the Discussion section and in the Supplementary text.*”

Figure 2: Caption – when describing the model, the authors state that the vertical Z-axis is 8 km
and refer to Figure 2. However, in the figure caption you indicate that the model dimensions are
64x32x6 km (also, it should be km³). For consistency, please modify the caption.

Thanks pointing this typo out, it is now changed accordingly.

Reviewer #3 (Remarks to the Author):

This is a well written and presented manuscript using numerical modelling to show how differing amounts of magmatism may control the morphology of transform faults and fracture zones. This morphology can be some of the most variable and extreme anywhere on the ocean floor (perhaps on all of Earth), and so these exciting results presenting a possible mechanism for explaining some of this variability are significant and interesting to a wide variety of Earth scientists. Generally, I think that the manuscript is in a good shape. My main comments are that the narrative is perhaps oversimplified in a few different areas and there are a few oversights that should be addressed. The numerical modelling is outside of my area of expertise.

1. Some of the most extreme morphology with large ridges, often known as transverse ridges, at transform faults and fracture zones has been seen at places such as St Paul, Romanche, Chain, Vema, or at least is mostly seen at slow-spreading ridges. Most commonly, much of this topography has been thought to be due to transpression/tension due to non-orthogonal plate movements (e.g. Maia et al. 2016; Marjanovic et al. 2020). Many of these areas (e.g. Romanche) have been thought to have very low levels of magmatism, though this is now in debate with some evidence for more magmatism than previously thought (e.g. Gregory et al., 2021).

Thanks for pointing out this issue, which was also raised by Reviewer 1 and 2. As highlighted above, instead of trying to explain the complex morphology of individual transform faults, we seek to investigate the spreading-rate dependent broad-scale transform topography. We have revised our text to clarify this.

The authors should comment on these features at slow spreading ridges as they are different to the Mode 1 ridges produced in the modelling – in these locations is magmatism likely also the primary driver? Even though, for example, the St Paul rocks have been shown to consist of peridotites and mylonites (Maia et al., 2016)? Or, if there are plate tectonic directional controls, do these override the original magmatic controls on morphology? This is an interesting question that should at least be addressed.

As described in our responses above, we acknowledge the original description of “ridge” for Mode 1 case is confusing because there are different terms for ridge-like structures in the literature, e.g., ‘median ridge’, ‘transverse ridge’, (Maia, 2019); ‘positive flower structure’ (Harmon et al., 2018). Moreover, in some cases these ridge-like structures exist within the transform valley at intermediate and slow spreading systems (Supplementary Fig. 1).

To address this issue, we revised our description for the fast-spreading cases to clarify that our focus is on the overall trend of broad-scale topography rather than individual features:

“At the fastest seafloor spreading rates, the transform zone is relatively shallow, sometimes even shallower than the adjacent seafloor (Mode 1, Fig. 1a, Supplementary Fig. 1).”

Further we have added a discussion of both median and transverse ridges to the revised text. Specifically, median ridges that appear within transform valleys are observed across all spreading regimes. Their origins remain controversial and are typically described on a case-dependent basis (Maia, 2019). However, as we now discuss more clearly in the added paragraph in the Discussion section and as shown in added examples in Supplementary Fig. 1, most of the time (St. Paul is an outlier), their appearance does not overwrite the overall trend of average broad-scale topography of transform fault, which remains strongly a function of spreading rate.

Transverse ridges (as pointed out by the reviewer) are typically associated with trans-tension
due to non-orthogonal plate motions. In this study, we have focused only on the topography
that is generated by steady-state plate motion parallel to the transform zone. Thus, we have
explicitly ignored complexity such as transpression or transform normal extension due to
kinematic plate motion changes, which have been attributed to formation of large, elevated
ridges that bound the transform zone. We now explicitly point out that these complexities are not
addressed by our models.

As we added in the revised Discussion, the relative importance of local (Maia et al., 2016) vs.
far-field causes (Pockalny et al., 1997) of plate boundary reorganization and the origin of these
prominent features (e.g., transverse ridges) are worthy of further investigation.

Along with the tectonic controls, there are also potential effects on topography from the
composition of the lithosphere, and the age-offset relationship proposed by Ren et al. (2022)
which are not discussed, which is an oversight. Magmatism may be an important or even
primary driver, but that doesn't mean these other mechanisms become insignificant or have no
effect.

We have added a discussion on this point in the revised Introduction (citation names are written
below for clarity, but numbered in the manuscript):

*"More recently, Grevemeyer et al., (2021) proposed that the deepening of transform valleys is*
*due to age offset dependent tectonic thinning, which is caused by the increasing obliquity of the*
*transform plate boundary at greater depths. However, this model does not simulate transform*
*topography and uses surface kinematic boundary condition (Behn et al., 2007; Roland et al.,*
*2010), which are more appropriate for investigating viscous asthenosphere flow, but not so ideal*
*for addressing strain partitioning in the shallow brittle lithosphere. Relocated seismicity at the*
*fast spreading Gofar transform fault (Froment et al., 2014; Gong & Fan, 2022) and slow*
*spreading Chain transform fault (Schlaphorst et al., 2023) do not show evidence of increasing*
*obliquity of the plate boundary at greater depth, potentially indicating that these changes occur*
*beneath the brittle-ductile transition."*

and at the Discussion near the end of the main text:

*"Finally, a recent systematic survey of transform fault topography argued that age-offset is a*
*more important control on transform depth than spreading rate (Ren et al. 2022). However,*
*because there is a correlation between spreading rate and age offset (Luo et al. 2021), future*
*work will be needed to distinguish between these two effects."*

2. Please add a more detailed explanation as to why fracture zones are shallower than
transform faults in the model as these mechanisms weren't clear to me, and this is a significant
and interesting result that warrants a proper explanation. It is mentioned (line 194) that there is
less extension there as it is stronger, but doesn't the already deep transform fault then become
a fracture zone? So how does it become more shallow? The <1% asymmetry in dike intrusion is
not enough to raise seafloor depth significantly so cannot be the explanation in the model.
Please clarify in the manuscript or if you do not know the mechanisms then state this explicitly.

We appreciate this request. We have tried to simplify this discussion in the revised manuscript
by clarifying the roles of shear-driven tension and dike opening in generating the difference in
transform fault and fracture zone topography. Specifically, we revised the text and added

Supplementary Fig. 4 to illustrate how materials are accreted and transported near the
transform domain dike zone and how these patterns also contribute to the depth difference.

*“For most cases except Mode 1 models with the highest M_T , the transform valleys are*
*consistently deeper than their adjacent fracture zones (Fig. 4a). The reason for this is shear-*
*driven tension only affects the active transform, not the fracture zones. Another contributing*
*factor for the depth difference is that the shear resistance of transform fault causes incomplete*
*lateral opening of the transform dike, which in turn promotes the intruded material to migrate*
*upward (Supplementary Fig. 4). The result is to produce shallower topography in the fracture*
*zone as the new material is accreted and advected away from the active transform fault. Note*
*that while we do not model lava eruptions, this mechanism supports the conceptual idea that*
*magma accretion in the transform domain is important in leading to shallower fracture zones*
*(Grevemeyer et al., (2021)³.”*

3. How possible or likely is it to have $M_t > 1$? Is there precedent for this in other studies of
magmatic vs tectonic spreading at mid-ocean ridges as I assumed the maximum value of M was
1? I am also wondering about the likelihood of having more magmatism in the transform domain
than along the main ridge segment ($M_t > M$). Is this relative relationship (M_t vs M) significant – i.e.
higher or lower rates of magmatism within the transform domain when compared to the ridge,
rather than the M_t value alone? Please add something to justify these values.

As described in our original Line 94-98, our $M_T = 1$ case corresponds to $M=0.5$ at each end of
the transform zone. So even for our maximum M_T value of 1.1, it is equivalent to M value of 0.55
as in previous studies that use the M factor for a single mid-ocean ridge segment (e.g., Behn &
Ito, 2008; Buck et al., 2005; J.-A. Olive et al., 2015; Tian & Choi, 2017).

We have further clarified this in the text by adding the sentence below to right after the original
Line 98:

*“The fractional rate of extension due to diking at each end of the transform fault is $M_T/2$, such*
*that M_T represents the combined opening rate of the two transform dike zones. So, when $M_T =$*
*1, the transform zone dike accretion rate for each segment corresponds to half of the plate*
*separation rate.”*

4. Generally, this study is modelling transform faults as hybrid extensional-transform plate
boundaries, rather than being solely strike-slip, with full spreading occurring in the transform
zone. This is only mentioned in the last sentence of the abstract but is a significant feature of
the study and should be addressed in the discussion section also.

Agreed. We added the sentence below following the original Line 198 to clarify this point:

*“In this mode, the transform fault is a hybrid extensional and strike-slip plate boundary.”*

5. I found Extended data Figure 3 very helpful in understanding the mechanisms for forming the
topography in the different modes. It would be better to have this figure in the main text.

As Nature Communication allows up to 10 display items, we now added it to the main text as
Figure 5.

6. Relating to the sensitivity tests done on water overburden, these are explained in a
misleading way (paragraph starting line 205; line 492; line 561) which suggests that multiple

water depths have been tested. My understanding from the text is that only no water and 2 km
of water have been tested – thus what has been tested is the uncertainty in the model results for
not taking into account the density of water, not changes in seafloor depth. Please change all
parts of the manuscript/supplement where this is explained to clarify this, or better, run at least
one more test with a different seafloor depth (e.g. 3 km) with water density included.

As suggested, we ran a series of additional models with seafloor depth of 4 km, now shown in
Supplementary Fig. 5 and in Supplementary Table 2. These runs confirm that our model results
are not sensitive to water depth, whether it is 2 km, 4 km or without water overburden.

We choose to not include the water overburden because it saves computational time.
The fact that water depth does not affect the results allows us to compare results with transform
faults at various water depths ranging between 2 to 4 km.

7. Relating to the other sensitivity tests done – these are good and are a beneficial part of the
study to help understand the effects of these different parameters on the TF/FZ morphology. I
found the way the results are explained/presented not ideal in the main manuscript and the
supplementary text – it would be better to have these variations also plotted as error bars on the
model data in figures 3a and 4a please. Then they can be compared by eye to the variations in
the real data in 3b and 4b.

We thought about this suggestion which could help with explaining the larger scatter for slow
spreading systems shown in the data because the variations from sensitivity test are largest for
the Mode 3 case. However, because the overall variability is approximately contained within the
spread of points in a given value of M_T , and because Figures 3 & 4 are already busy plots with
multiple dimensions of information, we feel adding these variations to the plot will make it
distracting for the overall trends that we are focusing on.

We instead prefer to show those sensitivity variations with Supplementary Fig. 5, 6 and Table 2.
In addition, we added to the caption of Figures 3 & 4 to clarify this point by stating “*Note that*
*variability due to other parameters is tested and is approximately contained within the spread of*
*points for a given value of M_T (See Supplementary Information for details).*”

Also, in sentences starting lines 212 and 215, I wouldn't agree that changes on the order of
'only' hundreds of metres are insignificant or mean the predictions are 'insensitive' (though I
realise they are less than this for two of the modes). It would be better and more useful to the
reader to present these results/parameters as meaningful in some scenarios and perhaps
responsible for some of the other variation seen in observed data, even if secondary to the
effect of magmatism. This comment relates both to the section in the main manuscript and how
the sensitivity test results are presented and discussed in the supplement.

Thanks for the suggestion. We made the following adjustment to the sentences from original
L212-215:

“*Variations in model results due to different half spreading rates within the global range (1, 2, or*
*5 cm/yr), as well as a factor of ~2 difference in lithospheric thickness (4, 6 or 8 km) or transform*
*fault length (32 or 62 km) are negligible for Mode 1 & 2, but are larger (a few hundred meters)*
*for Mode 3 (Supplementary Fig. 6). These variations can partially explain the larger scatter*
*seen in the observations at slower spreading rates (Fig. 3e and Fig. 4b). Meanwhile, we*
*hypothesize that the greater variability in magma supply at slower spreading rates¹⁸ may also*
*be partially responsible for the larger scatter in the natural data. Overall, we find that over this*
*range of parameter space, predicted transform and fracture zone depths cluster closely along*
*the overall trend of the reference model results (Fig. 3&4). We therefore conclude that our*

model results point to M_T as being a globally prominent control on the broad-scale averaged
topography of oceanic transform faults.”

Please see below when overlapping the sensitivity test results supplementary Fig. 6 onto our
Fig 3d and Fig. 4a to support the above sentence that “predicted transform and fracture zone
depths cluster closely along the overall trend of the reference model results (Figs. 3 & 4)”.

I'm not sure I agree there is much support for the hypothesis on line 223 – some of this scatter
could be related to the tested parameters, as well as other things that can affect TF/FZ
topography such as non-orthogonal far-field plate tectonic stresses.

We rephrased the sentence to:

“Meanwhile, we hypothesize that the greater variability in magma supply at slower spreading
rates (Jean-Arthur Olive & Dublanche, 2020) may also be partially responsible for the larger
scatter in the natural data.”

Please also see the added sentence: “... but are larger (a few hundred meters) for Mode 3
cases (Supplementary Fig. 6). These variations can partially explain the larger scatter seen in
the observations at slower spreading rates (Fig. 3e and Fig. 4b)...”

8. In general, there are more grammatical errors and the text is less clear in the Supplementary
section. Please go over and edit this more carefully.

Done.

Other comments by line

Line 53: I find this section explaining that only two mechanisms are invoked a little
oversimplified, and ignore the recent studies of Grevemeyer et al. (2021) and Ren et al. (2022),
aside from a small comment about the shallowing of fracture zones. These recent papers and
their findings should be included in more detail in this section.

Following the suggestion, we added more detailed descriptions of Grevemeyer et al. (2021) in
the Introduction and Ren et al. (2022) in the Discussion before the end of the main text.

Line 57-58: Are there examples/references for this claim for transform ridges?

See our reply to comment 1 above.
More specifically, Maia (2019) and reference therein indicate that many of the plate motion
changes last a few million years so they are ‘not persistent in time’. We envision that whenever
there is a kinematic plate motion change, as described in Maia (2019), depending on a few
factors (transform length, age offset, spreading rates etc.), sooner or later within a few million
568 years, the transform fault should respond to the plate motion change and reach a new steady-
569 state that their broad-scale averaged topography is to the first-order controlled by the long-term
averaged magmatism. Plate motion change clearly cannot explain the global spreading rate
dependent transform fault topography.

(Harmon et al., 2018) suggest the variations of median ridges (positive flower structure) at the
Chain transform fault are due to temporal variations in magmatism and ridge propagation rather
than plate-scale readjustments. Meanwhile the overall broad-scale valley for Chain transform is
not due to plate motion change.

Line 68-69: Have there been any seismic studies that can confirm this idea of an increase in
crustal thickness at fast-spreading transforms? Studies I am aware of (e.g. at Gofar & Quebrada
on the EPR, Roland et al (2012)) don’t report thicker crust, so please add ‘potentially’ or similar.
Or add some more references for support here.

We revised the sentence to:
*“In contrast, gravity data at fast-spreading mid-ocean ridges have been used to infer thicker*
*crust along transform faults (Gregg et al., 2007), indicating enhanced magmatism.”*

Line 131 (Figure 3): Why weren’t all strength and width combinations tested? E.g. widths of 2
588 km at $M_t > 1.02$.

As described in original Line 98-102:
*“The transform domain width is set to be either 1 km (for fast spreading cases) or 2 km (for*
*intermediate to slow spreading cases), which is roughly consistent with a recent bathymetric*
*analyses³⁴ that yielded a global median transform width of 2.5 km, but a slightly smaller median*
*width of 1.8 km for systems with a full spreading rate >8 cm/yr.”*

We choose to use narrower transform width for fast spreading cases to reflect the trend of
narrower transform width with faster spreading rate that has been shown in a few recent studies
(e.g., Luo et al., 2021; Ren et al., 2022).

Also, given that the y axis is variable in (d), please add some small tick marks with a regular
interval (e.g. every 0.05), so this variability/non-linearity is clear.

Changed accordingly.

We also note this change in the figure caption at original Line 142-145:
*“In (d), the vertical axis height of M_T from 0.70 to 0.98 (Mode 3) is scaled by 1/7 (indicated by*
*tick marks with a constant interval of 0.4) so as to....”*

Line 139-141: Sentence beginning ‘Negative values’ needs rewording, I take it you mean the top
axis is difference in depth from the initial surface, and the bottom axis is the absolute depth.

We revised the text to:
*“Negative values along the top axis implies subsidence relative to the initial model surface at 0*
*km, whereas larger values in the bottom axis means deeper seafloor for comparing with data.”*

Line 165: Sentence beginning ‘Compared to Mode 2’. Please slightly reword this sentence for

clarity, e.g. 'Mode 2 valleys depths are more sensitive to transform strength than Mode 3,
increasing in depth with increasing transform strength', or another wording of your choice.

Following suggestion, we revised the sentence to:

"*Compared to Mode 2 models, results of transform valley depth for Mode 3 models are less*
*sensitive to the transform strength (Supplementary Fig. 3)*"

Line 232: The reported asymmetry is less than 1% - I don't see how that is significant enough to
make a measurable impact on crustal thickness. Please either elaborate on this e.g. could
higher percentages of asymmetry be possible for particular reasons, or change/remove this.

Please see reply above on the same point when discussing why fracture zones are shallower.

Methods/supplementary sections

Line 404: The choice of 1 and 2 km wide transforms is confusing – most transforms in the
reference quoted, Ren et al (2022), are wider than 2 km (average 3.7 km) and why only look at
those with a spreading rate >8 cm/year? Ren doesn't include fast-spreading transforms – if this
is why then please explain in the text. Also, why is a median quoted as a range, shouldn't this
be a single value? It would have been interesting to see the effects of wider transforms, as is
more common at slow spreading rates.

Our original text might be misleading: the "2.5 to 1.8 km" is not intended to mean a range but for
respective median values of 2.5 km for all cases and 1.8 km for cases with spreading rate
higher than 8 cm/yr. 2.5 km and 1.8 km are calculated using the supplementary data file
provided by Ren et al., 2022. We revised it to the following:

"*The transform domain width is set to be either 1 km (for fast spreading cases) or 2 km (for*
*intermediate to slow spreading cases), which is roughly consistent with a recent bathymetric*
*analyses³⁴ that yielded a global median transform width of 2.5 km, but a slightly smaller median*
*width of 1.8 km for systems with a full spreading rate >8 cm/yr.*"

Note that the Luo et al. (2021) analysis show higher values of transform width (on the order of
10 km), which is due to their different definition of valley width. Our choice here is aimed to
address the global trend shown in Grevenmeyer 2021, while being roughly consistent with Ren et
al. (2022).

Line 451: Change to 'time 1 (t1) and time 2 (t2)' so the reader knows what these are in the
figure.

We revised both the Figure 5 and its caption regarding this point. We now annotate with points
A₀B₀ and A₁B₁ to simplify this illustration.

Line 461: Sentence starting 'With water' is confusing and the numbers given don't all match
those in Extended Data 2, which I'm guessing they should? Please rewrite to clarify and correct
the numbers, e.g. 'shallower and deeper' are the wrong way around for Mode 1.

Thanks for pointing this issue out. We now remove the confusing sentence and refer the reader
to the Supplementary Table 2 for details.

Line 469: Please change 'stroke colour' to 'outline' throughout this caption.

We now remove all the stroke/outline colors and use only different symbol shapes to
differentiate different models.

Line 507: Add 'observed' before data – as it needs to be clear you are comparing to the real
observed data.

Changed accordingly.

Lines 497 and 520: Please give a value in parentheses for these 'negligible changes' e.g. (<10
668 m).

We use 'negligible changes can be identified' under the context of global trend shown in
Supplementary Fig. 6. All results cluster near the base model shown as rectangles. We added
"(Supplementary Fig. 6 and Table 2)." at the end of the sentence to refer the reader to the
sources. Please also see above plot where Supplementary Fig. 6 and Fig. 3d and Fig. 4a are
overlapped to show all data collapses into one overall trend.

Line 526: Would be useful to add an explanation as to why these changes in spreading rate
have this effect on the depths.

This is a good point of future investigations.

Line 529 and line 544: Sentences starting 'Also, these variations' and 'The changes are also...'.
Please remove these sentences - if the trends are the same, I don't think this is relevant to
state.

We think these sentences are important because they explicitly mention that the model
uncertainties due to variations in parameters other than M_T are less than the measurement
uncertainties, which could help highlight the first-order control of M_T on the observables.

References:

Gregory, E. P. M., Singh, S. C., Marjanović, M., & Wang, Z. (n.d.). Evidence for thick mafic crust at the
slow-slipping Romanche oceanic transform fault. *Geology*. 2021. <https://doi.org/10.1130/G49097.1>

Grevemeyer, I., Rüpke, L. H., Morgan, J. P., Iyer, K., & Devey, C. W. (2021). Extensional tectonics and
two-stage crustal accretion at oceanic transform faults. *Nature*, [https://doi.org/10.1038/s41586-021-](https://doi.org/10.1038/s41586-021-03278-9)
[03278-9](https://doi.org/10.1038/s41586-021-03278-9)

Maia, M., Sichel, S., Briais, A., Brunelli, D., Ligi, M., Ferreira, N., Campos, T., Mougél, B., Brehme, I.,
Hémond, C., Motoki, A., Moura, D., Scalabrin, C., Pessanha, I., Alves, E., Ayres, A., & Oliveira, P.
(2016). Extreme mantle uplift and exhumation along a transpressive transform fault. *Nature*
*Geoscience*, <https://doi.org/10.1038/ngeo2759>

Marjanović, M., Singh, S. C., Gregory, E. P. M., Grevemeyer, I., Growe, K., Wang, Z., Vaddineni, V.,
Laurencin, M., Carton, H., Gómez de la Peña, L., & Filbrandt, C. (2020). Seismic Crustal Structure and
Morphotectonic Features Associated With the Chain Fracture Zone and Their Role in the Evolution of the
Equatorial Atlantic Region. *Journal of Geophysical Research: Solid*
*Earth*, <https://doi.org/10.1029/2020jb020275>

Ren, Y., Geersen, J., & Grevemeyer, I. (2022). Impact of Spreading Rate and Age-Offset on Oceanic
Transform Fault Morphology. *Geophysical Research Letters*, <https://doi.org/10.1029/2021GL096170>

Roland, E., Lizarralde, D., McGuire, J. J., & Collins, J. A. (2012). Seismic velocity constraints on the
material properties that control earthquake behavior at the Quebrada-Discovery-Gofar transform faults,
East Pacific Rise. *Journal of Geophysical Research: Solid Earth* <https://doi.org/10.1029/2012JB009422>

References:

Behn, M. D., & Ito, G. (2008). Magmatic and tectonic extension at mid-ocean ridges: 1. Controls on fault

characteristics. *Geochemistry, Geophysics, Geosystems*, 9(8).

Behn, M. D., Boettcher, M. S., & Hirth, G. (2007). Thermal structure of oceanic transform faults. *Geology*, 35(4),
307–310.

Bonatti, E., Brunelli, D., Buck, W. R., Cipriani, A., Fabretti, P., Ferrante, V., et al. (2005). Flexural uplift of a
lithospheric slab near the Vema transform (Central Atlantic): Timing and mechanisms. *Earth and Planetary*
*Science Letters*, 240(3–4), 642–655.

Buck, W. R., Lavier, L. L., & Poliakov, A. N. (2005). Modes of faulting at mid-ocean ridges. *Nature*, 434(7034), 719–
723.

Cramer, F. (2018). Scientific colour maps. *Zenodo*, 10. <https://doi.org/10.5281/zenodo.1243862>

Cramer, F., Shephard, G. E., & Heron, P. J. (2020). The misuse of colour in science communication. *Nature*
*Communications*, 11(1), 5444.

Froment, B., McGuire, J. J., Van Der Hilst, R. D., Gouédard, P., Roland, E. C., Zhang, H., & Collins, J. A. (2014). Imaging
along-strike variations in mechanical properties of the Gofar transform fault, East Pacific Rise. *Journal of*
*Geophysical Research: Solid Earth*, 119(9), 7175–7194. <https://doi.org/10.1002/2014JB011270>

Gong, J., & Fan, W. (2022). Seismicity, fault architecture, and slip mode of the westernmost Gofar transform fault.
*Journal of Geophysical Research: Solid Earth*, 127(11), e2022JB024918.

Gregg, P. M., Lin, J., Behn, M. D., & Montési, L. G. J. (2007). Spreading rate dependence of gravity anomalies along
oceanic transform faults. *Nature*, 448(7150), 183–187. <https://doi.org/10.1038/nature05962>

Grevenmeyer, I., Rüpke, L. H., Morgan, J. P., Iyer, K., & Devey, C. W. (2021). Extensional tectonics and two-stage
crustal accretion at oceanic transform faults. *Nature*, 591(7850), 402–407.
<https://doi.org/10.1038/s41586-021-03278-9>

Harmon, N., Rychert, C., Agius, M., Tharimena, S., Le Bas, T., Kendall, J. M., & Constable, S. (2018). Marine
geophysical investigation of the Chain Fracture Zone in the equatorial Atlantic from the PI-LAB experiment.
*Journal of Geophysical Research: Solid Earth*, 123(12), 11–016.

Luo, Y., Lin, J., Zhang, F., & Wei, M. (2021). Spreading rate dependence of morphological characteristics in global
oceanic transform faults. *Acta Oceanologica Sinica*, 40(4), 39–64.

Maia, M. (2019). Topographic and morphologic evidences of deformation at oceanic transform faults: far-field and
local-field stresses. In *Transform plate boundaries and fracture zones* (pp. 61–87). Elsevier.

Maia, M., Sichel, S., Briais, A., Brunelli, D., Ligi, M., Ferreira, N., et al. (2016). Extreme mantle uplift and exhumation
along a transpressive transform fault. *Nature Geoscience*, 9(8), 619–623.

Olive, J.-A., Behn, M. D., Ito, G., Buck, W. R., Escartín, J., & Howell, S. (2015). Sensitivity of seafloor bathymetry to
climate-driven fluctuations in mid-ocean ridge magma supply. *Science*, 350(6258), 310–313.

Olive, Jean-Arthur, & Dublanchet, P. (2020). Controls on the magmatic fraction of extension at mid-ocean ridges.
*Earth and Planetary Science Letters*, 549, 116541.

Pockalny, R. A., Fox, P. J., Fornari, D. J., Macdonald, K. C., & Perfit, M. R. (1997). Tectonic reconstruction of the
Clipperton and Siqueiros Fracture Zones: Evidence and consequences of plate motion change for the last 3
750 Myr. *Journal of Geophysical Research: Solid Earth*, 102(B2), 3167–3181.
<https://doi.org/10.1029/96JB03391>

Ren, Y., Geersen, J., & Grevemeyer, I. (2022). Impact of Spreading Rate and Age-Offset on Oceanic Transform Fault
Morphology. *Geophysical Research Letters*, 49(2), e2021GL096170.

Roland, E., Behn, M. D., & Hirth, G. (2010). Thermal-mechanical behavior of oceanic transform faults: Implications
for the spatial distribution of seismicity: THERMOMECHANICAL BEHAVIOR OF OCEANIC TRANSFORM
FAULTS. *Geochemistry, Geophysics, Geosystems*, 11(7), n/a-n/a. <https://doi.org/10.1029/2010GC003034>

Schlaphorst, D., Rychert, C. A., Harmon, N., Hicks, S. P., Bogiatzis, P., Kendall, J.-M., & Abercrombie, R. E. (2023).
Local seismicity around the Chain Transform Fault at the Mid-Atlantic Ridge from OBS observations.
*Geophysical Journal International*, ggad124.

Tian, X., & Choi, E. (2017). Effects of axially variable diking rates on faulting at slow spreading mid-ocean ridges.
*Earth and Planetary Science Letters*, 458, 14–21. <https://doi.org/10.1016/j.epsl.2016.10.033>

Tucholke, B. E., & Schouten, H. (1988). Kane fracture zone. *Marine Geophysical Researches*, 10, 1–39.

Wolfson-Schwehr, M., & Boettcher, M. S. (2019). Global characteristics of oceanic transform fault structure and
seismicity. In *Transform plate boundaries and fracture zones* (pp. 21–59). Elsevier. Retrieved from
<https://www.sciencedirect.com/science/article/pii/B9780128120644000025>

REVIEWERS' COMMENTS

Reviewer #2 (Remarks to the Author):

Review for manuscript: Magmatism Controls Global Oceanic Transform Fault Topography by Tian et al.

The authors addressed all my concerns and comments by modifying the text and figures accordingly. I have no additional remarks and recommend the manuscript for publication.

Reviewer #3 (Remarks to the Author):

This is the second time reviewing this manuscript after the authors have made their revisions from the previous round.

I feel satisfied that the authors have addressed the concerns raised and have no further comments. This is a well presented paper and a very interesting study suggesting a primary mechanism for the observed variability in transform fault and fracture zone morphology, which would be of interest to the wide readership of Nature Communications.

Thank you for your efforts - I enjoyed reading the paper!